# A GENERAL THEORY OF RELATIVITY IN REINFORCEMENT LEARNING

## ABSTRACT

We propose a new general theory measuring the relativity between two arbitrary Markov Decision Processes (MDPs) from the perspective of reinforcement learning (RL). Considering two MDPs, tasks such as policy transfer, dynamics modeling, environment design, and simulation to reality (sim2real), etc., are all closely related. The proposed theory deeply investigates the connection between any two cumulative expected returns defined on different policies and environment dynamics, and the theoretical results suggest two new general algorithms referred to as Relative Policy Optimization (RPO) and Relative Transition Optimization (RTO), which can offer fast policy transfer and dynamics modeling. RPO updates the policy using the *relative policy gradient* to transfer the policy evaluated in one environment to maximize the return in another, while RTO updates the parameterized dynamics model (if there exists) using the *relative transition gradient* to reduce the gap between the dynamics of the two environments. Then, integrating the two algorithms offers the complete algorithm Relative Policy-Transition Optimization (RPTO), in which the policy interacts with the two environments simultaneously, such that data collections from the two environments, policy and transition updates are all completed in a closed loop to form a principled learning framework for policy transfer. We demonstrate the effectiveness of RPO, RTO and RPTO in the OpenAI gym's classic control tasks by creating policy transfer problems.

## 1 INTRODUCTION

Deep reinforcement learning (RL) has demonstrated its great successes in recent years, including breakthrough of solving a number of challenging problems like Atari (Mnih et al., 2015), GO (Silver et al., 2016; 2017), DOTA2 (Berner et al., 2019) and StarCraft II (Vinyals et al., 2019), with human-level performance or even beyond. These successes demonstrate that current deep RL methods are capable to explore and exploit sufficiently in huge observation and action spaces, as long as sufficient and effective data samples can be generated for training, such as the cases in games. For example, AlphaGo Zero (Silver et al., 2017) costs 3 days of training over 4.9 millions self-play games, and OpenAI Five (Berner et al., 2019) and AlphaStar (Vinyals et al., 2019) spend months of training using thousands of GPUs/TPUs over billions of generated matches. However, for environments that prohibit infinite interactions, e.g., robotics, real life traffic control and autopilot, etc., applying general RL is difficult because generating data is extremely expensive and slow. Even if parallel data collection is possible, for example, by deploying multiple robots or vehicles running simultaneously, the scale of collected data is still far below that in virtual games. Worse still, exploration in these environments is considerably limited for safety reasons, which further reduces the effectiveness of the generated data. Due to the above challenges, similar significant advances like in solving virtual games have not been witnessed in these applications yet.

Generally, there are three tracks of approaches targeting to alleviate the aforementioned situation to promote the widespread application of RL. They are improving the data efficiency, transfer learning and simulator engineering.

To improve data efficiency, many recent efforts have been paid on investigating offline RL algorithms (Siegel et al., 2020; Fujimoto et al., 2019; Kumar et al., 2019; Wu et al., 2019; Wang et al., 2018). Compared to standard on-policy or off-policy RL, offline RL (also known as batch RL) aims

to effectively use previously collected experiences stored in a given dataset, like supervised learning, without online interactions with the environment. The stored experiences may not be generated by a fixed or known policy, so offline RL algorithms can leverage any previously collected data and learn a provably better policy than those who generated the experiences in the dataset. Although offline RL can effectively take advantage of finite data samples, solving a complex real-world task still requires huge amount of high quality offline experiences. Another way to increase data efficiency is to adopt model-based RL. Compared to model-free methods, model-based RL (Kaiser et al., 2019; Janner et al., 2019; Moerland et al., 2020) learns a dynamics model that mimics the transitions in the true environment, and then the policy can feel free to interact with the learned dynamics instead of the true environment. It has been proved that the true return can be improved by interacting with the learned dynamics model when the model error is bounded (Janner et al., 2019). However, learning an accurate dynamics model still requires sufficient transition data by interacting with the true environment, specifically for complex dynamics with noisy transitions.

Transfer learning in RL (Zhu et al., 2020) is practically useful to adapt a policy learned in a *source* environment to solve another task in the *target* environment. In the context of this paper, we consider the case where the policy can feel free to explore in the source environment, while the amount of collected data in the target environment should be as small as possible. When the source environment is a simulated one while the target environment takes place in reality, the transfer problem is also known as the simulation to reality (sim2real) problem. The simplest way to do transfer is to train the policy in the source environment and then use the converged parameters as warm start for a new policy or part of its parameters in the target environment, so that the amount of interactions with the target is expected to be largely reduced, as long as the tasks and dynamics in the two environments are closely related. Training a shared or partially shared policy in both the source and target environments is an alternative method which also belongs to the multi-task reinforcement learning scope (Hessel et al., 2019). Domain adaptation has been demonstrated to be another useful technique (Ibarz et al., 2021). Such methods try to bridge the gap between the source and target environments using some adaptation networks. For example, adapter networks were introduced to convert the input in simulation to be close to the real-world observation, by utilizing the generative adversarial model (James et al., 2017; Shrivastava et al., 2017; Bousmalis et al., 2017; 2018; Rao et al., 2020), or on the contrary an inverse network was trained to convert real-world observation to that in simulation (James et al., 2019). Using such adapter networks, the policy only needs to be trained in the source environment, and then it can directly be applied in the target environment.

An important concept in transfer learning is that instead of directly deploying RL in the target environment, a source environment is considered as a proxy. Sharing this spirit, the last track of approaches tries to build a proxy simulator that is as close as possible to the target environment, and hence we refer to such methods as simulator engineering. For example, in robotics control problems, there are many mature toolboxes can offer simulation engineering, including MuJoCo, PyBullet, Gazebo, etc. Model-based RL can also be viewed as a specific form of simulator engineering that the simulator is composed by a pure neural network, which is trained to approach the target environment as with lower model error as possible, while this might require a large amount of dynamics data in the target environment as mentioned above. Actually, to achieve more efficient and accurate simulator engineering, one recent rising direction is to integrate differentiable programming and physical systems to build a trainable simulator, which follows the physical laws as in reality and also whose key factors, such as the mass, length and friction of some objects, are trainable like the parameters in neural networks. Representative examples include the DiffTaichi (Hu et al., 2020), Brax (Freeman et al., 2021) and Nimble (Werling et al., 2021).

Overall, the existing methods focus on either directly improving the data efficiency in the target environment or bridging/reducing the gap between a proxy environment and the target environment, and there lacks a principled theory that can incorporate the learning in the two environments through an unified framework, and explain the intrinsic relationship between the expected returns in the two environments from the perspective of RL. In this paper, we inherit the spirit in transfer learning and consider two environments, where one is free to interact and another is the goal to solve, and the number of interactions in the goal environment should be as small as possible. We believe that there exist some explicit connections between the expected returns in the two environments, given two different policies, from the very fundamental perspective of RL. To verify this, we formally define two Markov Decision Processes (MDPs) and then explicitly derive the difference between the expected returns in the two MDPs. In the following context, with the RL convention, an environment

is equivalent to an MDP. Specifically, let $\mathcal{P}(s'|s, a)$ and $\mathcal{P}'(s'|s, a)$ denote two dynamics transition functions in any two arbitrary MDPs sharing the same state and action spaces, where $(s, a, s')$ is the tuple of the state, action and next state. Let $\pi'(a|s)$ and $\pi(a|s)$ denote two arbitrary policies, and denote $J(\mathcal{P}, \pi)$ as the cumulative expected return given $\mathcal{P}$ and $\pi$. Then, we aim to investigate the difference $J(\mathcal{P}', \pi) - J(\mathcal{P}, \pi')$, which is referred to as the *relativity gap* between the two MDPs. It turns out that the relativity gap has a very interesting and compact form that integrates the interactions in both environments. Now, suppose $\mathcal{P}$ and $\mathcal{P}'$ are the dynamics functions in the source and target MDPs respectively, and $J(\mathcal{P}, \pi')$ has been maximized by optimizing $\pi'$. Then, with fixed $\mathcal{P}$, $\mathcal{P}'$ and $\pi'$, maximizing the relativity gap over $\pi$ by constraining $\pi$ to be close to $\pi'$ will also improve the return $J(\mathcal{P}', \pi)$ in the target MDP; on the other hand, for trainable $\mathcal{P}$, minimizing the relativity gap by optimizing $\mathcal{P}$ given a fixed policy $\pi = \pi'$ will reduce the dynamics gap, similar to what is done by conventional model-based RL methods. Based on the above two principles, our theoretical results suggest two general algorithms referred to as Relative Policy Optimization (RPO) and Relative Transition Optimization (RTO), respectively. RPO updates the policy using the *relative policy gradient* to transfer the policy evaluated in the source environment to maximize the return in the target environment, while RTO updates a dynamics model using the *relative transition gradient* to reduce the value gap in the two environments. Then, applying RPO and RTO simultaneously offers a complete algorithm named Relative Policy-Transition Optimization (RPTO), which can transfer the policy from the source to the target smoothly. RPO, RTO and RPTO interact with the two environments simultaneously, so that data collections from two environments, policy and/or transition updates are completed in a closed loop to form a principled learning framework. In the experimental section, we show how to practically apply RPO, RTO and RPTO algorithms. We demonstrate the effectiveness of these methods in the classic control problems in OpenAI gym with both discrete and continuous actions, by varying the physical variables like mass, length and gravity of the objects to create policy transfer problems. At the last section, we discuss a few new directions based on the proposed relativity theory, which are worthy future investigations.

## 2 PRELIMINARIES

### 2.1 REINFORCEMENT LEARNING

A standard RL problem can be described by a tuple $\langle \mathcal{E}, \mathcal{A}, \mathcal{S}, \mathcal{P}, r, \gamma, \pi \rangle$, where $\mathcal{E}$ indicates the environment that is an MDP with dynamics transition probability $\mathcal{P}$; at each time step $t$, $s_t \in \mathcal{S}$ is the global state in the state space $\mathcal{S}$, and $a_t \in \mathcal{A}$ is the action executed by the agent at time step $t$ from the action space $\mathcal{A}$; the dynamics transition function $\mathcal{P}(s_{t+1}|s_t, a_t)$ is the probability of the state transition $(s_t, a_t) \rightarrow s_{t+1}$; for the most general case, the reward $r(s_t, a_t, s_{t+1})$ can be written as a function of $s_t, a_t$ and $s_{t+1}$, while in many tasks it only relies on one or two of them, or it is even a constant in sparse rewards problem. For notation simplicity, we usually write $r(s_t, a_t, s_{t+1})$ as $r_t$; $\gamma \in [0, 1]$ is a discount factor and $\pi(a_t|s_t)$ denotes a stochastic policy. The following equations define some important quantities in reinforcement learning. The objective of RL is to maximize the expected discounted return

$$J(\mathcal{P}, \pi) = \mathbb{E}_{s_0, a_0, \cdots \sim \mathcal{P}, \pi} \left[ \sum_{t=0}^{\infty} \gamma^t r_t \right], \text{ where } s_0 \sim \mathcal{P}(s_0),\ a_t \sim \pi(a_t|s_t),\ s_{t+1} \sim \mathcal{P}(s_{t+1}|s_t, a_t).$$

At time step $t$, the state-action value $Q^{\mathcal{P}, \pi}$, value function $V^{\mathcal{P}, \pi}$, and advantage $A^{\mathcal{P}, \pi}$ are defined as $Q^{\mathcal{P}, \pi}(s_t, a_t) = \mathbb{E}_{s_{t+1}, a_{t+1}, \cdots \sim \mathcal{P}, \pi} \left[ \sum_{l=0}^{\infty} \gamma^l r_{t+l} \right]$, $V^{\mathcal{P}, \pi}(s_t) = \mathbb{E}_{a_t, s_{t+1}, \cdots \sim \mathcal{P}, \pi} \left[ \sum_{l=0}^{\infty} \gamma^l r_{t+l} \right]$, $A^{\mathcal{P}, \pi}(s, a) = Q^{\mathcal{P}, \pi}(s, a) - V^{\mathcal{P}, \pi}(s)$. In the above standard definitions, we explicitly show their dependence on both the dynamics $\mathcal{P}$ and policy $\pi$, since we will analyze these functions defined on variant dynamics and policies. This convention will be kept throughout the paper.

### 2.2 TRPO AND PPO

Given two arbitrary policies $\pi'$ and $\pi$, a well-known policy improvement theorem (Kakade & Langford, 2002; Schulman et al., 2015) is the fact revealed by the following equation

$$J(\mathcal{P}, \pi') = J(\mathcal{P}, \pi) + \mathbb{E}_{s_0, a_0, \cdots \sim \mathcal{P}, \pi'} \left[ \sum_{t=0}^{\infty} \gamma^t A^{\mathcal{P}, \pi}(s_t, a_t) \right]. \tag{1}$$

Based on this theorem, some widely adopted RL algorithms such as TRPO (Schulman et al., 2015) and PPO (Schulman et al., 2017) are developed. In TRPO, the following objective is optimized

$$\text{maximize}_\theta \; \mathbb{E}_{s\sim d^{\mathcal{P},\pi_{\theta_{old}}}, a\sim\pi_{\theta_{old}}} \left[ \frac{\pi_\theta(a|s)}{\pi_{\theta_{old}}(a|s)} A^{\mathcal{P},\pi_{\theta_{old}}}(s,a) \right],$$

$$\text{subject to } \mathbb{E}\left[ \mathcal{KL}\left( \pi_{\theta_{old}}(\cdot|s) || \pi_\theta(\cdot|s) \right) \right] \le \delta,$$

where the policy is parameterized by $\theta$, and $\theta_{old}$ is the parameter since last update. $d^{\mathcal{P},\pi}$ is the discounted visitation probability given $\mathcal{P}$ and $\pi$. $\mathcal{KL}(\cdot||\cdot)$ is the KL-divergence and $\delta$ is a small constant restricting the policy update step size. To simplify the optimization, PPO removes the constraint in TRPO and maximizes the following clipped version

$$\mathbb{E}_{s\sim d^{\mathcal{P},\pi_{\theta_{old}}}, a\sim\pi_{\theta_{old}}} \left[ \min\left( R(\theta) A^{\mathcal{P},\pi_{\theta_{old}}}(s,a), \text{clip}\left( R(\theta), 1-\epsilon, 1+\epsilon \right) A^{\mathcal{P},\pi_{\theta_{old}}}(s,a) \right) \right],$$

where $R(\theta) = \frac{\pi_\theta(a|s)}{\pi_{\theta_{old}}(a|s)}$ and $\epsilon$ is a hyperparameter controlling the proportion of clipped data. We review the TRPO and PPO algorithms here to allow readers to conveniently compare and see the connection between them and our proposed algorithms in the following sections.

## 3 THE THEORY OF RELATIVITY IN RL

With awareness of the most fundamental RL definitions in Section 2.1, it is sufficient to proceed the theory in this section. To be straightforward, we directly give the most important theorem revealing the theory of relativity in RL below. All the proofs are provided in the appendix.

**Theorem 1 (The Theory of Relativity in RL)** *Given two Markov Decision Processes (MDPs) denoted by $\mathcal{E}'$ and $\mathcal{E}$, who share the same state and action spaces $\mathcal{S}$ and $\mathcal{A}$, their dynamics transition probabilities are $\mathcal{P}'(s_{t+1}|s_t, a_t)$ and $\mathcal{P}(s_{t+1}|s_t, a_t)$ for any transition $(s_t, a_t) \to s_{t+1}$ in $\mathcal{E}'$ and $\mathcal{E}$, respectively. Assume the initial state distributes in the two MDPs identically that $\mathcal{P}'(s_0) = \mathcal{P}(s_0)$. Let $J(\mathcal{P},\pi)$ denote the expected return defined on dynamics $\mathcal{P}$ and policy $\pi$. Then, the relativity gap between any two expected returns under different dynamics and policies is defined as*

$$\underbrace{J(\mathcal{P}',\pi) - J(\mathcal{P},\pi')}_{\text{relativity gap}} = \underbrace{J(\mathcal{P}',\pi) - J(\mathcal{P},\pi)}_{\text{dynamics-induced gap}} + \underbrace{J(\mathcal{P},\pi) - J(\mathcal{P},\pi')}_{\text{policy-induced gap}}, \tag{2}$$

*such that the dynamics-induced gap has an explicit form as*

$$J(\mathcal{P}',\pi) - J(\mathcal{P},\pi) = \mathbb{E}_{s_0,a_0,\cdots\sim\mathcal{P}',\pi} \sum_{t=0}^\infty \gamma^t \left[ r(s_t, a_t, s_{t+1}) + \gamma V^{\mathcal{P},\pi}(s_{t+1}) - Q^{\mathcal{P},\pi}(s_t, a_t) \right], \tag{3}$$

*and the policy-induced gap is revealed by the policy improvement theorem (Kakade & Langford, 2002) introduced in Eq. (1), rewritten by inverting $\pi$ and $\pi'$ as*

$$J(\mathcal{P},\pi) - J(\mathcal{P},\pi') = \mathbb{E}_{s_0,a_0,\cdots\sim\mathcal{P},\pi} \sum_{t=0}^\infty \gamma^t A^{\mathcal{P},\pi'}(s_t, a_t). \tag{4}$$

In Theorem 1, it is surprising that the dynamics-induced gap has a very compact formulation like the policy-induced gap. Below, we emphasize a few important points implied in Eqs. (3) and (4):

- In Eq. (3), the expectation is taken over the trajectory $s_0, a_0, \cdots$ sampled from $(\mathcal{P}', \pi)$, while the value and state-action value functions in the expectation, i.e., $V^{\mathcal{P},\pi}(s_{t+1})$ and $Q^{\mathcal{P},\pi}(s_t, a_t)$, are value functions defined on $(\mathcal{P}, \pi)$. This reveals a very practically useful conclusion that given a fixed policy $\pi$, the dynamics-induced gap can be calculated by measuring the value functions $V^{\mathcal{P},\pi}(s_{t+1})$ and $Q^{\mathcal{P},\pi}(s_t, a_t)$ in the dynamics $\mathcal{P}$ (imagining this is the source environment, where infinite data can be generated to accurately evaluate the value functions), while collecting probably a few data samples in $\mathcal{P}'$ (imagining this is the target environment) to estimate the expectation.

- In Eq. (3), $\mathbb{E}_{s_{t+1}}[r(s_t, a_t, s_{t+1}) + \gamma V^{\mathcal{P},\pi}(s_{t+1})] \ne Q^{\mathcal{P},\pi}(s_t, a_t)$, because the transition $(s_t, a_t) \to s_{t+1}$ takes place in $\mathcal{P}'$ instead of $\mathcal{P}$, and hence Eq. (3) is not zero whereas $\mathcal{P}'(s_{t+1}|s_t, a_t) \ne \mathcal{P}(s_{t+1}|s_t, a_t)$ happens for non-zero $r(s_t, a_t, s_{t+1})$ and $V^{\mathcal{P},\pi}(s_{t+1})$ with high probability, especially for high-dimensional deep neural networks. Of course, if $\mathcal{P}' = \mathcal{P}$, we immediately have $J(\mathcal{P}',\pi) = J(\mathcal{P},\pi)$ from Eq. (3).

- The dynamics-induced gap, i.e., $J(\mathcal{P}', \pi) - J(\mathcal{P}, \pi)$, is related to what was analyzed in model-based policy optimization (MBPO) (Janner et al., 2019). Unfortunately, MBPO bounds the value gap with the dynamics model error at very early derivations to get their main theorem, which provides a basic theoretical support for that the commonly adopted model-based RL algorithms (alternatively updating the model and policy) can guarantee policy improvement, while it does not suggest new algorithms. Instead, with much deeper investigation of the fundamental dynamics-induced value gap, we can finally get the explicit identity equation in Eq. (3), instead of a bound. As we will show later, Theorem 1 suggests two thoroughly new algorithms for policy transfer and transition update. Details of the superiority of Theorem 1 over MBPO can be found in the proofs in appendix.

So far, Theorem 1 provides important results on both dynamics-induced and policy-induced gaps, while it is still not clear how these results can be applied empirically. In the following sections, we will introduce two new practical algorithms derived from Theorem 1, where one algorithm is for fast policy transfer from the source environment to the target environment, and another algorithm updates the parameterized dynamics in the source environment to be close to the dynamics in the target environment. Then, by combining the two algorithms, we obtain the complete algorithm to fast transfer a policy from source to target.

## 4 RELATIVE POLICY OPTIMIZATION (RPO)

As discussed previously, Eq. (3) in Theorem 1 suggests a way of estimating the dynamics-induced value gap by evaluating $Q^{\mathcal{P},\pi}(s_t, a_t)$ and $V^{\mathcal{P},\pi}(s_{t+1})$ in $\mathcal{P}$ while sampling data in $\mathcal{P}'$, given $\pi$. Practically, it is of less interest to estimate the exact dynamics-induced gap. Instead, if we have trained a policy $\pi^*$ in $\mathcal{P}$ (source environment) that maximizes $J(\mathcal{P}, \pi)$, then we are interested in finding another $\hat{\pi}$ such that $\hat{\pi} = \arg\max_\pi [J(\mathcal{P}', \pi) - J(\mathcal{P}, \pi^*)]$, i.e., $\hat{\pi}$ maximizes the dynamics-induced gap, and also indirectly maximizes $J(\mathcal{P}', \pi)$. Normally, as long as $\mathcal{P}'$ is not far from $\mathcal{P}$, finding $\hat{\pi}$ can use $\pi^*$ as a warm start. Based on the above motivation, we propose the following theorem to get a lower bound of the dynamics-induced value gap.

**Theorem 2** *Define $D_{TV}^{max}(p, q) = \max_x D_{TV}(p(\cdot|x)||q(\cdot|x))$ as the total variation divergence between two distributions $p(\cdot|x)$ and $q(\cdot|x)$, where $D_{TV}(p(\cdot|x)||q(\cdot|x)) = \frac{1}{2} \sum_y |p(y|x) - q(y|x)|$. Define $\epsilon = \max_{s,a} |A^{\mathcal{P},\pi}(s, a)|$, where $A^{\mathcal{P},\pi}(s, a) = Q^{\mathcal{P},\pi}(s, a) - V^{\mathcal{P},\pi}(s)$ is the advantage. Let*

$$\delta_1 = D_{TV}^{max}(\mathcal{P}'(\cdot|s,a), \mathcal{P}(\cdot|s,a))$$

*for any $(s, a) \in \mathcal{S}, \mathcal{A}$ and let*

$$\delta_2 = D_{TV}^{max}(\pi'(\cdot|s), \pi(\cdot|s))$$

*for any $s \in \mathcal{S}$, and any two policies $\pi'$ and $\pi$. Let $r_{max} = \max_{s,a,s'} r(s, a, s')$ be the max reward for all $(s, a, s')$. Now, let $\Delta^{\mathcal{P}',\mathcal{P}}(\pi) = J(\mathcal{P}', \pi) - J(\mathcal{P}, \pi)$ denote the dynamics-induced gap as a function of $\mathcal{P}'$, $\mathcal{P}$ and $\pi$. Now, we import a new policy $\pi'$ and define the following function*

$$L_{\pi'}(\pi) = \sum_{t=0}^{\infty} \gamma^t \, \mathbb{E}_{s_0, a_0, \cdots, s_t \sim \mathcal{P}', \pi'} \sum_{a_t} \pi(a_t|s_t) \sum_{s_{t+1}} \mathcal{P}'(s_{t+1}|s_t, a_t)$$

$$\left[ r(s_t, a_t, s_{t+1}) + \gamma V^{\mathcal{P},\pi'}(s_{t+1}) - Q^{\mathcal{P},\pi'}(s_t, a_t) \right]$$

*as an approximation of $\Delta^{\mathcal{P}',\mathcal{P}}(\pi)$ by sampling $s_0, a_0, \cdots$ using $\pi'$ and evaluating $V^{\mathcal{P},\pi'}$ and $Q^{\mathcal{P},\pi'}$ using $\pi'$. Then, we have the following lower bound*

$$\Delta^{\mathcal{P}',\mathcal{P}}(\pi) \geq L_{\pi'}(\pi) - \frac{2\gamma \delta_1 \epsilon}{(1 - \gamma)^2} - \frac{4 r_{max} \delta_1 \delta_2 \gamma}{(1 - \gamma)^3}.$$

Based on Theorem 2, we can further obtain the following lower bound of the entire relativity gap.

**Proposition 1** *The entire relativity gap in Eq. (2) has the following lower bound*

$$J(\mathcal{P}', \pi) - J(\mathcal{P}, \pi') \geq \mathbb{E}_{s \sim d^{\mathcal{P}',\pi'}, a, s' \sim \mathcal{P}', \pi'} \frac{\pi(a|s)}{\pi'(a|s)} [r(s, a, s') + \gamma V^{\mathcal{P},\pi'}(s') - V^{\mathcal{P},\pi'}(s)] - C,$$

*where $s'$ is the next state that $(s, a) \to s'$, and $C = \frac{2\gamma\delta_1\epsilon + 4\epsilon\gamma\delta_2^2}{(1-\gamma)^2} + \frac{4r_{max}\delta_1\delta_2\gamma}{(1-\gamma)^3}$ is a constant relying on the dynamics difference $\delta_1$ and policy difference $\delta_2$.*

Now, it becomes clear that by taking $\pi = \pi_\theta$ and $\pi' = \pi_{\theta_{old}}$ for some policy parameters $\theta$ and its old version $\theta_{old}$ since last update, Proposition 1 suggests the following empirical objective

$$\text{maximize}_\theta \; \mathbb{E}_{s \sim d^{\mathcal{P}', \pi_{\theta_{old}}}, a, s' \sim \mathcal{P}', \pi_{\theta_{old}}} \frac{\pi_\theta(a|s)}{\pi_{\theta_{old}}(a|s)} [r(s, a, s') + \gamma V^{\mathcal{P}, \pi_{\theta_{old}}}(s') - V^{\mathcal{P}, \pi_{\theta_{old}}}(s)],$$

$$\text{subject to } \mathbb{E}_{s \sim d^{P', \pi_{\theta_{old}}}} [D_{TV}(\pi_{\theta_{old}}(\cdot|s) || \pi_\theta(\cdot|s))] \leq \varepsilon, \tag{5}$$

for some small $\varepsilon$. At the first glance, the objective in Eq. (5) is very similar to the standard RL problem considered in TRPO or PPO (Schulman et al., 2015; 2017). However, again, by noting where the data is sampled from and how $V$ and $Q$ values are evaluated, we can realize the important difference that in Eq. (5), $r(s, a, s') + \gamma V^{\mathcal{P}, \pi_{\theta_{old}}}(s') - V^{\mathcal{P}, \pi_{\theta_{old}}}(s) \neq A^{\mathcal{P}, \pi_{\theta_{old}}}(s, a)$, i.e., it is not the general advantage function, because the transition $(s, a) \to s'$ takes place in $\mathcal{P}'$. To be more accurate, we call $r(s, a, s') + \gamma V^{\mathcal{P}, \pi_{\theta_{old}}}(s') - V^{\mathcal{P}, \pi_{\theta_{old}}}(s)$ the *Relative Advantage*.

That is, Eq. (5) suggests an empirical algorithm that we sample $(s, a, s')$ from $\mathcal{E}'$ using $\pi_{old}$ and compute the values $V^{\mathcal{P}, \pi_{old}}(s')$ and $V^{\mathcal{P}, \pi_{\theta_{old}}}(s)$ using $\pi_{\theta_{old}}$ in $\mathcal{E}$ to update $\theta$. Therefore, we refer to the optimization of Eq. (5) as the Relative Policy Optimization (RPO) algorithm.

The objective in Eq. (5) contains a constraint on the policy update size, and directly solving this objective requires line search, similar to what was proposed in TRPO (Schulman et al., 2015). To simplify the optimization, we use a clipped version, which has been demonstrated to be effective in PPO (Schulman et al., 2017).

Proposition 1 implies that as long as we can improve Eq. (5) by at least the constant $C$, we can guarantee improvement on $J(\mathcal{P}', \pi_\theta)$ over the constant $J(\mathcal{P}, \pi_{\theta_{old}})$ at the current step. Generally, RPO optimizes $\theta$ to maximize the relativity gap $J(\mathcal{P}', \pi_\theta) - J(\mathcal{P}, \pi_{\theta_{old}})$ in the direction of maximizing $J(\mathcal{P}', \pi_\theta)$, because $J(\mathcal{P}, \pi_{\theta_{old}})$ is a constant at the current step. However, starting from a well trained policy $\pi_{\theta^*}$ in $\mathcal{P}$, as RPO updates $\theta$, $\pi_{\theta_{old}}$ will be gradually far from $\pi_{\theta^*}$, and therefore $J(\mathcal{P}, \pi_{\theta_{old}})$ might decrease. If this happens, continuing maximizing the relativity gap $J(\mathcal{P}', \pi_\theta) - J(\mathcal{P}, \pi_{\theta_{old}})$ can not guarantee the increase of $J(\mathcal{P}', \pi_\theta)$. One possible solution is to optimize the RPO loss plus the standard RL loss, e.g., the PPO loss, defined on $\mathcal{P}$ to keep $\pi_\theta$ always performing well in $\mathcal{E}$. As we can imagine, in such case, $\pi_\theta$ will be updated towards a robust policy that performs well in both $\mathcal{E}'$ and $\mathcal{E}$. Indeed, as we will show in our experiments, as long as $\mathcal{P}'$ is not too far away from $\mathcal{P}$, optimizing RPO + PPO is often able to obtain such a robust policy; however, once $\mathcal{P}'$ differs from $\mathcal{P}$ too much, RPO will fail to transfer the policy to the target environment. This gives a hint that in addition to RPO, we need further to reduce the gap between $\mathcal{P}'$ and $\mathcal{P}$, if $\mathcal{P}'$ and $\mathcal{P}$ are far away from each other. This is possible when $\mathcal{P}$ (the dynamics in source environment) is trainable, which has been considered in physical dynamics modeling (Hu et al., 2020; Freeman et al., 2021; Werling et al., 2021) and model-based RL methods (Janner et al., 2019).

## 5 RELATIVE TRANSITION OPTIMIZATION (RTO)

In this section, given fixed $\pi$ and $\mathcal{P}'$, we consider a trainable $\mathcal{P}$. Suppose $\mathcal{P}_\phi(s'|s, a)$ is parameterized by $\phi$ for any transition $(s, a) \to s'$. In the following theorem, we will import three dynamics quantities $\mathcal{P}'$, $\mathcal{P}_\phi$ and $\mathcal{P}_{\phi'}$, where $\mathcal{P}'$ still can be imagined as the dynamics in a target environment, and $\mathcal{P}_\phi$ and $\mathcal{P}_{\phi'}$ are two variant dynamics functions parameterized by $\phi$ and $\phi'$, respectively.

**Theorem 3** *Using the definitions in Theorem 2, the following function*

$$L_{\phi'}(\phi) = \sum_{t=0}^{\infty} \gamma^t \mathbb{E}_{s_0, a_0, \cdots, a_t \sim \mathcal{P}', \pi} \left[ \mathbb{E}_{s_{t+1} \sim \mathcal{P}'} \left[ r(s_t, a_t, s_{t+1}) + \gamma V^{\mathcal{P}_{\phi'}, \pi}(s_{t+1}) \right] - \right.$$

$$\left. \mathbb{E}_{s_{t+1} \sim \mathcal{P}_\phi} \left[ r(s_t, a_t, s_{t+1}) + \gamma V^{\mathcal{P}_{\phi'}, \pi}(s_{t+1}) \right] \right]$$

*is an approximation of $\Delta^{\mathcal{P}', \mathcal{P}_\phi}(\pi)$ by evaluating the value using $\mathcal{P}_{\phi'}$ instead of $\mathcal{P}_\phi$, and we have*

$$|\Delta^{\mathcal{P}', \mathcal{P}_\phi}(\pi) - L_{\phi'}(\phi)| \leq \frac{4\delta_1^2 \gamma r_{max}}{(1-\gamma)^3}.$$

In order to reduce the dynamics-induced gap by updating $\phi$, we have to minimize $|\Delta^{\mathcal{P}',\mathcal{P}_\phi}(\pi)|$. Based on Theorem 3, we have $|\Delta^{\mathcal{P}',\mathcal{P}_\phi}(\pi)| \leq |L_{\phi'}(\phi)| + \frac{4\delta_1^2 \gamma r_{max}}{(1-\gamma)^3}$. Therefore, we can alternatively minimize $|L_{\phi'}(\phi)|$. Empirically, if we consider $\mathcal{P}_\phi$ as a probability function, then by noting that

$$L_{\phi'}(\phi) = \sum_{t=0}^{\infty} \gamma^t \, \mathbb{E}_{s_0,\cdots,a_t \sim \mathcal{P}',\pi} \sum_{s_{t+1}} \left( \mathcal{P}'(s_{t+1}|s_t,a_t) - \mathcal{P}_\phi(s_{t+1}|s_t,a_t) \right) \left( r(s_t,a_t,s_{t+1}) + \gamma V^{\mathcal{P}_{\phi'},\pi}(s_{t+1}) \right),$$

and taking $\phi' = \phi_{old}$, minimizing $|L_{\phi'}(\phi)|$ is equivalent to optimizing the following least square

$$\text{minimize}_\phi \, \mathbb{E}_{s \sim d^{\mathcal{P}',\pi}, a \sim \pi} \sum_{s'} \left( \mathcal{P}'(s'|s,a) - \mathcal{P}_\phi(s'|s,a) \right)^2 \left( r(s,a,s') + \gamma V^{\mathcal{P}_{\phi_{old}},\pi}(s') \right)^2. \quad (6)$$

That is, we sample $(s,a)$ in $\mathcal{E}'$ using $\pi$ and optimize Eq. (6). We refer to the above optimization as Relative Transition Optimization (RTO). It is easy to see that RTO implies the standard model-based RL methods, who directly train $\phi$ using supervised learning, i.e., $\text{minimize}_\phi \, \mathbb{E}_{s \sim d^{\mathcal{P}',\pi}, a \sim \pi} \sum_{s'} \left( \mathcal{P}'(s'|s,a) - \mathcal{P}_\phi(s'|s,a) \right)^2$. Indeed, the objective of RTO in Eq. (6) can be viewed as a weighted form of supervised learning, with the weight $r(s,a,s') + \gamma V^{\mathcal{P}_{\phi_{old}},\pi}(s')$ showing that transitions happened with larger values evaluated by $\mathcal{P}_{\phi_{old}}$ and $\pi$ should be optimized more aggressively. This is fairly reasonable from the perspective of fitting values, instead of fitting the dynamics directly in supervised learning. Therefore, RTO provides a theoretical explanation for the standard model-based RL methods by reducing the dynamics-induced value gap, and absorbs standard model-based RL as a special case in RTO.

In practice, the transition function $\mathcal{P}_\phi$ is usually treated as a deterministic model, i.e., $\hat{s}' = \phi(s,a)$. Under such case, RTO optimizes

$$\text{minimize}_\phi \, \mathbb{E}_{\substack{s \sim d^{\mathcal{P}',\pi}, \\ a, s' \sim \mathcal{P}',\pi}} \left( r(s,a,s') + \gamma V^{\mathcal{P}_{\phi_{old}},\pi}(s') - r(s,a,\phi(s,a)) - \gamma V^{\mathcal{P}_{\phi_{old}},\pi}(\phi(s,a)) \right)^2, \quad (7)$$

and the common model-based methods solve $\text{minimize}_\phi \, \mathbb{E}_{s \sim d^{\mathcal{P}',\pi}, a, s' \sim \mathcal{P}',\pi} \left( s' - \phi(s,a) \right)^2$. Comparing the two objectives, Eq. (7) is more general by noting that fitting value function can distinguish the importance of states, and it is not necessary to learn a perfect dynamics model over all transitions.

In the experiments, we use deterministic dynamics model. More precisely, we adopt deterministic physical dynamics model, similar to recent proposed differentialable simulators (Hu et al., 2020; Freeman et al., 2021; Werling et al., 2021), which take advantage of the physical processes. Such model is much more efficient than pure neural network based dynamics model, because only a few scalars such as mass, length, gravity, etc., are trainable parameters in the physical systems.

## 6 RELATIVE POLICY-TRANSITION OPTIMIZATION (RPTO) ALGORITHM

Combining RPO and RTO algorithms, we finally obtain the complete Relative Policy-Transition Optimization (RPTO) algorithm in Algorithm 1. As we observe in Algorithm 1, in the main loop of

---

**Algorithm 1:** Relative Policy-Transition Optimization (RPTO)

---

1. Give the source and target environments $\mathcal{E}^{source}$ and $\mathcal{E}^{target}$, and their dynamics $\mathcal{P}_\phi^{source}$ and $\mathcal{P}^{target}$, where the source dynamics $\mathcal{P}_\phi^{source}$ is parameterized by $\phi$; give a well-trained policy $\pi_{\theta_0}$ in $\mathcal{E}_{\phi_0}^{source}$, where $\phi_0$ perfectly describes the initial source dynamics;
2. Create two empty replay buffers $\mathcal{D}_{source}$ and $\mathcal{D}_{target}$;
3. Initialize $\theta = \theta_0$ and $\phi = \phi_0$;
**while** *True* **do**
    4. Using $\pi_\theta$ to interact with $\mathcal{E}_\phi^{source}$ and push the generated trajectories into $\mathcal{D}_{source}$;
    5. Using $\pi_\theta$ to interact with $\mathcal{E}^{target}$ and push the generated trajectories into $\mathcal{D}_{target}$;
    6. Sample a mini-batch $\{(s,a,s')\}_{source} \sim \mathcal{D}_{source}$, and update $V^{\mathcal{P}_\phi^{source},\pi_\theta}$ by minimizing the TD-error;
    7. Sample a mini-batch $\{(s,a,s')\}_{target} \sim \mathcal{D}_{target}$, and apply the relative policy gradient in RPO to update $\pi_\theta$; at the same time, update $\mathcal{P}_\phi^{source}$ according to RTO;
**end while**

---

Table 1: An overview of the studied pairs of environments.

| CartPole-v0 | | | | MountainCarContinuous-v0 | | |
|---|---|---|---|---|---|---|
| | Source Env | Target Env | | | Source Env | Target Env |
| Pole Length | 0.5 | 0.6 | Gravity | | 0.0025 | 0.006 |
| Acrobot-v1 | | | | Pendulum-v0 | | |
| | Source Env | Target Env | | | Source Env | Target Env |
| Link Mass1 | 1.0 | 0.1 | Gravity | | 10.0 (Earth) | 3.72 (Mars) |
| Link Mass2 | 1.0 | 0.1 | | | | |

RPTO, the policy $\pi_\theta$ interacts with the two environments simultaneously and pushes the data into two buffers separately. In step 6, we sample a mini-batch from $\mathcal{D}_{source}$ to update the value function in Eq. (5) from RPO. In step 7, we sample a mini-batch from $\mathcal{D}_{target}$ to update the policy parameter $\theta$ according to Eq. (5) in RPO, and also update the dynamics $\phi$ according to Eqs. (6) or (7) from RTO. Therefore, RPTO combines data collection from two environments, RPO and RTO in one closed loop, and this offers a principled learning framework for policy transfer. Indeed, steps 6 and 7 can be parallelized, as long as the value function $V_\mu^{\mathcal{P}_\phi^{source}, \pi_\theta}$ (suppose it is parameterized by $\mu$) does not share parameters with $\pi_\theta$, i.e., $\mu$ and $\theta$ are independent. Also, $\mu$ can be updated more frequently because it only requires data that can be generated infinitely from the source environment, and the more accurate the value function in the source environment is, the more accurate the relative policy gradient in step 7 will be estimated. For the dynamics parameter $\phi$, the case becomes different, and it is often not a good choice to update $\phi$ as fast as we can, for example, using a larger step size for $\phi$. To understand this, we need to explain how RPTO transfers the policy to the target environment in advance.

As we can imagine, as $\mathcal{P}_\phi^{source}$ approaches $\mathcal{P}^{target}$ gradually by updating $\phi$ in step 7, the policy $\pi_\theta$ is able to interact with a sequence of smoothly varying source environments. During this training period, the policy sees much more diverse transitions that lie between the initial source environment (with dynamics $\mathcal{P}_{\phi_0}^{source}$) and the target environment. These diverse transitions are very helpful to encourage the agent to explore a robust policy. On the other hand, since the policy is initialized with a well-trained $\theta_0$ in $\mathcal{P}_{\phi_0}^{source}$, smoothly varying $\mathcal{P}_{\phi_0}^{source}$ to reach $\mathcal{P}^{target}$ generates a sequence of environments that naturally provide a curriculum learning scheme, and this is very similar to what is considered in the recently emerged Environment Design methods (Dennis et al., 2020). Now, we are ready to answer the question in the last paragraph that why $\phi$ should not be updated aggressively: a slowly and smoothly varying $\phi$ provides more chances for the agent to see diverse transitions. Actually, if $\mathcal{P}_\phi^{source}$ approaches $\mathcal{P}_{target}$ too fast, RPTO will be similar to directly training PPO in the target environment with a warm start $\theta_0$. In our experiments, we simply keep the learning rate of $\phi$ the same as that in RPO.

## 7 EXPERIMENTS

We experiment with all the OpenAI gym's classic control tasks, including CartPole-v0, MountainCarContinuous-v0, Acrobot-v1, and Pendulum-v0, because in these tasks the physical systems are explicitly coded, which allows us to easily build physical dynamics model with only a few trainable factors, e.g., pole length, gravity and link mass, as indicated in Table 1. Among these, CartPole-v0 and Acrobot-v1 are of discrete action space and the other two are continuous control problems. For all the tasks, the default settings in OpenAI gym are treated as the source environments, while for each task, we arbitrarily modify some of its physical factors to create the corresponding target environments. Details of the source and target environments are shown in Table 1.

For all tasks, we first pre-train a converged policy with PPO in the source environment, and the pre-trained policy will be used as a warm start when transferring to the target environment. For the policy transfer stage, PPO with the pre-trained policy as initialization, denoted as PPO-warm, will always be used as the baseline method. Other implementation details, such as hyper-parameters and neural network structures, are provided in the appendix. All the experiments in this section are repeated 10 times to plot the mean curve with standard derivation region.

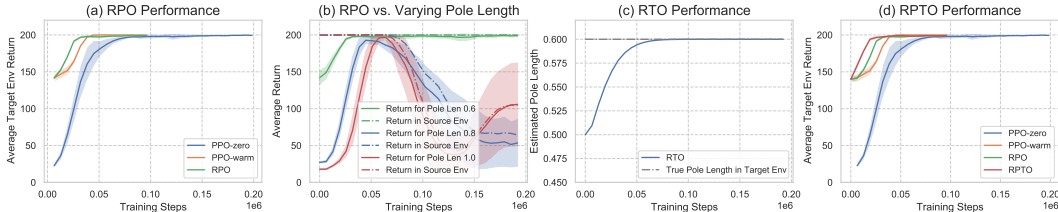

Figure 1: Tutorial experiments in CartPole-v0.

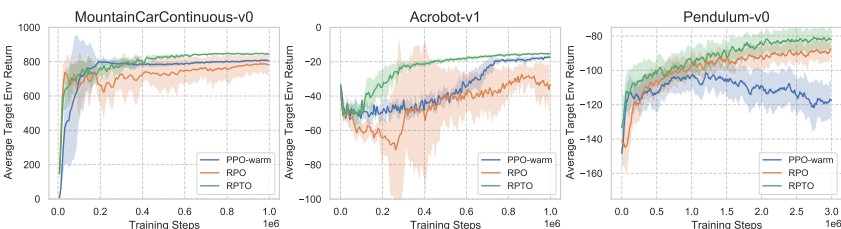

Figure 2: Overall performance in other tasks.

We first use CartPole-v0 as a tutorial environment to show how the proposed algorithms are practically applied. Fig. 1 reports the results for various evaluations of RPO, RTO and RPTO. Fig. 1(a) demonstrates that RPO (without RTO) is sufficient to successfully transfer the policy to the target environment with pole length of 0.6, and RPO transfers the policy faster than PPO-warm. In Fig. 1(b), by varying the pole length to test more target environments, we find that when the pole length difference becomes larger, RPO fails to obtain a stable policy in the target environment. The curves also imply that for the case where RPO succeeds, the learned policy can perform well in both the source and target environments, i.e., RPO finds a robust policy; while for the failed cases, the learned policy deteriorates in both environments either. These results are consistent with our analysis at the end of Section 4. Fig. 1(c) evaluates the performance of RTO. As we can observe, RTO can optimize $\phi$ to converge to the true pole length in the target environment. Finally, Fig. 1(d) reports the RPTO's performance, where the curves of PPO-zero, PPO-warm and RPO are duplicated from Fig. 1(a) as baselines. As we can observe, RPTO transfers the policy much faster than all the other methods.

So far, we have demonstrated how to apply RPO, RTO and RPTO in CartPole-v0 and what characteristics of these algorithms can help us better understand their performance. Now, we directly report all the performance curves for the other tasks in Fig. 2. In all the tasks, RPTO shows its superiority over PPO-warm in terms of both fast policy transfer and even better asymptotic convergence, because RPTO sees much more diverse dynamics that promotes exploration. Generally, RPO also shows its capability on fast policy transfer, while it usually suffers from higher variance and unstable performance, because it is limited by the dynamics gap between the source and target environments as revealed by Theorem 2 by noting the dynamics gap $\delta_1$ is a fixed constant in RPO. For the case in RPTO, benefitting from RTO, $\delta_1$ approaches zero gradually and so RPTO is stable and efficient.

## 8 DISCUSSION AND FUTURE WORK

We have proposed the general theory of relativity in RL. The theory shows its significance in creating new algorithms that are empirically demonstrated effective and efficient. In addition to RPO, RTO and RPTO, the relativity theory also opens a few new future directions in RL. For example, as we have discussed at the end of Section 6, controlling the update step size or frequency of RTO can provide better curriculum learning or environment design (although in this paper we simply fix the learning coefficient of RTO as the same as RPO. Please see appendix). Connecting this with meta-RL and current environment design algorithms is a promising future direction. Moreover, the theory and algorithms in this paper are orthogonal to other techniques commonly used in policy transfer, such as domain adaptation, domain randomization, augmented observation, etc., and all of these methods can be integrated together in policy transfer applications. It is of great interests to apply all these techniques to solve complex real-world problems like robotics in future research.

## REPRODUCIBILITY

According to the author guide, we provide a Reproducibility Statement here. For theoretical results in this paper, we have provided all complete proofs in the appendix. For the algorithms, we have attached the codes of RPO, RTO and RPTO in the submitted .zip supplementary material.

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

APPENDIX

## A  PROOF OF THEOREM 1

Define the expected cumulative return as

$$J(\mathcal{P}, \pi) = \mathbb{E}_{s_0, a_0, \cdots \sim \mathcal{P}, \pi} \left[ \sum_{t=0}^{\infty} \gamma^t r(s_t, a_t, s_{t+1}) \right],$$

where

$$s_0 \sim \rho(\mathcal{P}_0),\ a_t \sim \pi(a_t | s_t),\ s_{t+1} \sim \mathcal{P}(s_{t+1} | s_t, a_t).$$

To better understand the expected cumulative return, we have

$$
\begin{aligned}
J(\mathcal{P}, \pi) &= \sum_{t=0}^{\infty} \gamma^t \mathbb{E}_{s_t, a_t, s_{t+1}} r(s_t, a_t, s_{t+1}) \\
&= \sum_{t=0}^{\infty} \gamma^t \mathbb{E}_{s_t \sim p(s_t), a_t \sim \pi(a_t|s_t), s_{t+1} \sim \mathcal{P}(s_{t+1}|s_t, a_t)} r(s_t, a_t, s_{t+1}) \\
&= \sum_{t=0}^{\infty} \gamma^t \sum_{s_t} p(s_t) \sum_{a_t} \pi(a_t|s_t) \sum_{s_{t+1}} \mathcal{P}(s_{t+1}|s_t, a_t) r(s_t, a_t, s_{t+1}),
\end{aligned}
$$

where $p(s_t)$ indicates the probability that $s_t$ is visited. With a fixed policy $\pi$, we investigate the value difference between two MDPs as

$$
\begin{aligned}
&J(\mathcal{P}', \pi) - J(\mathcal{P}, \pi) \\
&= \sum_{t=0}^{\infty} \gamma^t \Big[ \mathbb{E}_{s_t \sim p'(s_t), a_t \sim \pi(a_t|s_t), s_{t+1} \sim \mathcal{P}'(s_{t+1}|s_t, a_t)} r(s_t, a_t, s_{t+1}) - \\
&\qquad\qquad \mathbb{E}_{s_t \sim p(s_t), s_{t+1} \sim \mathcal{P}(s_{t+1}|s_t, a_t), a_t \sim \pi(a_t|s_t)} r(s_t, a_t, s_{t+1}) \Big] \\
&= \sum_{t=0}^{\infty} \gamma^t \Big[ \sum_{s_t} p'(s_t) \sum_{a_t} \pi(a_t|s_t) \sum_{s_{t+1}} \left( \mathcal{P}'(s_{t+1}|s_t, a_t) - \mathcal{P}(s_{t+1}|s_t, a_t) \right) r(s_t, a_t, s_{t+1}) + \\
&\qquad\qquad \sum_{s_t} \left( p'(s_t) - p(s_t) \right) \sum_{a_t} \pi(a_t|s_t) \sum_{s_{t+1}} \mathcal{P}(s_{t+1}|s_t, a_t) r(s_t, a_t, s_{t+1}) \Big].
\end{aligned}
\tag{8}
$$

It is worth mentioning that MBPO (Janner et al., 2019) just stops here at Eq. (8) and starts to bound the transition difference $\mathcal{P}'(s_{t+1}|s_t, a_t) - \mathcal{P}(s_{t+1}|s_t, a_t)$ and the visitation difference $p'(s_t) - p(s_t)$ with the dynamics model error, which is similar to $\delta_1$ defined in our Theorem 2. Then, it obtains a lower bound of Eq. (8) to conclude the main theorem in MBPO. As we will show below, this is still far away to reach our main theoretical results in Theorem 1.

Continuing from Eq. (8), the first term in Eq. (8) derives as

$$
\begin{aligned}
&\sum_{s_t} p'(s_t) \sum_{a_t} \pi(a_t|s_t) \sum_{s_{t+1}} \left( \mathcal{P}'(s_{t+1}|s_t, a_t) - \mathcal{P}(s_{t+1}|s_t, a_t) \right) r(s_t, a_t, s_{t+1}) \\
&= \sum_{s_t} \frac{p'(s_t)}{p(s_t)} p(s_t) \sum_{a_t} \pi(a_t|s_t) \sum_{s_{t+1}} \frac{\left( \mathcal{P}'(s_{t+1}|s_t, a_t) - \mathcal{P}(s_{t+1}|s_t, a_t) \right)}{\mathcal{P}(s_{t+1}|s_t, a_t)} \mathcal{P}(s_{t+1}|s_t, a_t) r(s_t, a_t, s_{t+1}) \\
&= \mathbb{E}_{s_t \sim p(s_t), a_t \sim \pi(a_t|s_t), s_{t+1} \sim \mathcal{P}(s_{t+1}|s_t, a_t)} \frac{p'(s_t)}{p(s_t)} \frac{\mathcal{P}'(s_{t+1}|s_t, a_t) - \mathcal{P}(s_{t+1}|s_t, a_t)}{\mathcal{P}(s_{t+1}|s_t, a_t)} r(s_t, a_t, s_{t+1}) \\
&= \mathbb{E}_{s_t, a_t, s_{t+1} \sim p} \frac{p'(s_t)}{p(s_t)} \frac{\mathcal{P}'(s_{t+1}|s_t, a_t) - \mathcal{P}(s_{t+1}|s_t, a_t)}{\mathcal{P}(s_{t+1}|s_t, a_t)} r(s_t, a_t, s_{t+1}).
\end{aligned}
$$

In the second term of Eq. (8), the difference of the marginal probabilities $p'(s_t) - p(s_t)$ can be expanded similarly as

$$
\begin{aligned}
&p'(s_t) - p(s_t) \\
=& \mathbb{E}_{s_{t-1} \sim p'(s_{t-1}), a_{t-1} \sim \pi(a_{t-1}|s_{t-1})} \mathcal{P}'(s_t|s_{t-1}, a_{t-1}) - \\
& \mathbb{E}_{s_{t-1} \sim p(s_{t-1}), a_{t-1} \sim \pi(a_{t-1}|s_{t-1})} \mathcal{P}(s_t|s_{t-1}, a_{t-1}) \\
=& \sum_{s_{t-1}} p'(s_{t-1}) \sum_{a_{t-1}} \pi(a_{t-1}|s_{t-1}) \left( \mathcal{P}'(s_t|s_{t-1}, a_{t-1}) - \mathcal{P}(s_t|s_{t-1}, a_{t-1}) \right) + \\
& \sum_{s_{t-1}} (p'(s_{t-1}) - p(s_{t-1})) \sum_{a_{t-1}} \pi(a_{t-1}|s_{t-1}) \mathcal{P}(s_t|s_{t-1}, a_{t-1}) \\
=& \mathbb{E}_{s_{t-1} \sim p(s_{t-1}), a_{t-1} \sim \pi(a_{t-1}|s_{t-1})} \frac{p'(s_{t-1})}{p(s_{t-1})} \left( \mathcal{P}'(s_t|s_{t-1}, a_{t-1}) - \mathcal{P}(s_t|s_{t-1}, a_{t-1}) \right) + \\
& \sum_{s_{t-1}} (p'(s_{t-1}) - p(s_{t-1})) \mathbb{E}_{a_{t-1} \sim \mathcal{P}(a_{t_1}|s_{t-1})} \mathcal{P}(s_t|s_{t-1}, a_{t-1}).
\end{aligned}
\tag{9}
$$

Plugging Eq. (9) back into the term of Eq. (8), we have

$$
\begin{aligned}
& \sum_{s_t} (p'(s_t) - p(s_t)) \sum_{a_t} \pi(a_t|s_t) \sum_{s_{t+1}} \mathcal{P}(s_{t+1}|s_t, a_t) r(s_t, a_t, s_{t+1}) \\
=& \sum_{s_t} (p'(s_t) - p(s_t)) \mathbb{E}_{a_t \sim \pi(a_t|s_t), s_{t+1} \sim \mathcal{P}(s_{t+1}|s_t, a_t)} r(s_t, a_t, s_{t+1}) \\
=& \mathbb{E}_{s_{t-1}, a_{t-1}, s_t, a_t, s_{t+1} \sim \mathcal{P}, \pi} \frac{p'(s_{t-1})}{p(s_{t-1})} \frac{\mathcal{P}'(s_t|s_{t-1}, a_{t-1}) - \mathcal{P}(s_t|s_{t-1}, a_{t-1})}{\mathcal{P}(s_t|s_{t-1}, a_{t-1})} r(s_t, a_t, s_{t+1}) + \\
& \sum_{s_{t-1}} (p'(s_{t-1}) - p(s_{t-1})) \mathbb{E}_{a_{t-1}, s_t, a_t, s_{t+1}} r(s_t, a_t, s_{t+1}).
\end{aligned}
\tag{10}
$$

Note that the second term in Eq. (10) has an identical form as that in Eq. (8) by expanding prior states and actions. Therefore, by recursively expanding Eq. (10) backward, we finally have

$$
\begin{aligned}
& J(\mathcal{P}', \pi) - J(\mathcal{P}, \pi) \\
=& \sum_{t=0}^{\infty} \gamma^t \Bigg[ \mathbb{E}_{s_t, a_t, s_{t+1} \sim \mathcal{P}, \pi} \frac{p'(s_t)}{p(s_t)} \frac{\mathcal{P}'(s_{t+1}|s_t, a_t) - \mathcal{P}(s_{t+1}|s_t, a_t)}{\mathcal{P}(s_{t+1}|s_t, a_t)} r(s_t, a_t, s_{t+1}) + \\
& \mathbb{E}_{s_{t-1}, a_{t-1}, s_t, a_t, s_{t+1} \sim \mathcal{P}, \pi} \frac{p'(s_{t-1})}{p(s_{t-1})} \frac{\mathcal{P}'(s_t|s_{t-1}, a_{t-1}) - \mathcal{P}(s_t|s_{t-1}, a_{t-1})}{\mathcal{P}(s_t|s_{t-1}, a_{t-1})} r(s_t, a_t, s_{t+1}) + \\
& \cdots \cdots \Bigg] \\
=& \sum_{t=0}^{\infty} \mathbb{E}_{\tau \sim \mathcal{P}, \pi} \sum_{i=0}^{t} \gamma^t \frac{p'(s_i)}{p(s_i)} \frac{\mathcal{P}'(s_{i+1}|s_i, a_i) - \mathcal{P}(s_{i+1}|s_i, a_i)}{\mathcal{P}(s_{i+1}|s_i, a_i)} r(s_t, a_t, s_{t+1}) \\
=& \mathbb{E}_{\tau \sim \mathcal{P}, \pi} \sum_{t=0}^{\infty} \gamma^t r(s_t, a_t, s_{t+1}) \sum_{i=0}^{t} \frac{p'(s_i)}{p(s_i)} \frac{\mathcal{P}'(s_{i+1}|s_i, a_i) - \mathcal{P}(s_{i+1}|s_i, a_i)}{\mathcal{P}(s_{i+1}|s_i, a_i)},
\end{aligned}
\tag{11}
$$

where $\tau$ indicates the trajectory $s_0, a_0, \cdots$.

Let $x_t = \gamma^t r(s_t, a_t, s_{t+1})$ and $y_i = \frac{p'(s_i)}{p(s_i)} \frac{\mathcal{P}'(s_{i+1}|s_i,a_i) - \mathcal{P}(s_{i+1}|s_i,a_i)}{\mathcal{P}(s_{i+1}|s_i,a_i)}$, we have

$$
\sum_{t=0}^{\infty} x_t \sum_{i=0}^{t} y_i = x_0 y_0 + x_1(y_0 + y_1) + x_2(y_0 + y_1 + y_2) + \cdots
$$

$$
= y_0(x_0 + x_1 + x_2 + \cdots) + y_1(x_1 + x_2 + \cdots) + \cdots = \sum_{i=0}^{\infty} y_i \sum_{t=i}^{\infty} x_t,
$$

and $\sum_{t=i}^{\infty} x_t = \gamma^i \sum_{t=i}^{\infty} \gamma^{t-i} r(s_t, a_t, s_{t+1}) = \gamma^i R_i$, where $R_i = \sum_{t=i}^{\infty} \gamma^{t-i} r(s_t, a_t, s_{t+1})$ is the empirical cumulative future reward. Continuing from Eq. (11),

$$J(\mathcal{P}', \pi) - J(\mathcal{P}, \pi)$$

$$= \mathbb{E}_{\tau \sim \mathcal{P}, \pi} \sum_{t=0}^{\infty} \gamma^t R_t \frac{p'(s_t)}{p(s_t)} \frac{\mathcal{P}'(s_{t+1}|s_t, a_t) - \mathcal{P}(s_{t+1}|s_t, a_t)}{\mathcal{P}(s_{t+1}|s_t, a_t)}$$

$$= \sum_{t=0}^{\infty} \gamma^t \mathbb{E}_{\tau \sim \mathcal{P}, \pi} R_t \frac{p'(s_t)}{p(s_t)} \frac{\mathcal{P}'(s_{t+1}|s_t, a_t) - \mathcal{P}(s_{t+1}|s_t, a_t)}{\mathcal{P}(s_{t+1}|s_t, a_t)}$$

$$= \sum_{t=0}^{\infty} \gamma^t \mathbb{E}_{s_0, a_0, \cdots, s_{t+1} \sim \mathcal{P}, \pi} \frac{p'(s_t)}{p(s_t)} \frac{\mathcal{P}'(s_{t+1}|s_t, a_t) - \mathcal{P}(s_{t+1}|s_t, a_t)}{\mathcal{P}(s_{t+1}|s_t, a_t)} \mathbb{E}_{a_{t+1}, s_{t+2}, \cdots \sim \mathcal{P}, \pi} R_t$$

$$= \sum_{t=0}^{\infty} \gamma^t \mathbb{E}_{s_0, a_0, \cdots, s_{t+1} \sim \mathcal{P}, \pi} \frac{p'(s_t)}{p(s_t)} \frac{\mathcal{P}'(s_{t+1}|s_t, a_t) - \mathcal{P}(s_{t+1}|s_t, a_t)}{\mathcal{P}(s_{t+1}|s_t, a_t)} \mathbb{E}_{a_{t+1}, s_{t+2}, \cdots \sim \mathcal{P}, \pi} [r(s_t, a_t, s_{t+1}) + \gamma R_{t+1}]$$

$$= \sum_{t=0}^{\infty} \gamma^t \mathbb{E}_{s_0, a_0, \cdots, s_{t+1} \sim \mathcal{P}, \pi} \frac{p'(s_t)}{p(s_t)} \frac{\mathcal{P}'(s_{t+1}|s_t, a_t) - \mathcal{P}(s_{t+1}|s_t, a_t)}{\mathcal{P}(s_{t+1}|s_t, a_t)} \left( r(s_t, a_t, s_{t+1}) + \gamma V^{\mathcal{P}, \pi}(s_{t+1}) \right)$$

$$= \sum_{t=0}^{\infty} \gamma^t \mathbb{E}_{s_0, a_0, \cdots, s_t, a_t \sim \mathcal{P}', \pi} \mathbb{E}_{s_{t+1} \sim \mathcal{P}, \pi} \frac{\mathcal{P}'(s_{t+1}|s_t, a_t) - \mathcal{P}(s_{t+1}|s_t, a_t)}{\mathcal{P}(s_{t+1}|s_t, a_t)} \left( r(s_t, a_t, s_{t+1}) + \gamma V^{\mathcal{P}, \pi}(s_{t+1}) \right)$$

$$= \sum_{t=0}^{\infty} \gamma^t \left\{ \mathbb{E}_{s_0, a_0, \cdots, s_t, a_t, s_{t+1} \sim \mathcal{P}', \pi} [r(s_t, a_t, s_{t+1}) + \gamma V^{\mathcal{P}, \pi}(s_{t+1})] - \mathbb{E}_{s_0, a_0, \cdots, s_t, a_t \sim \mathcal{P}', \pi} Q^{\mathcal{P}, \pi}(s_t, a_t) \right\}$$

$$= \mathbb{E}_{\tau \sim \mathcal{P}', \pi} \sum_{t=0}^{\infty} \gamma^t \left[ r(s_t, a_t, s_{t+1}) + \gamma V^{\mathcal{P}, \pi}(s_{t+1}) - Q^{\mathcal{P}, \pi}(s_t, a_t) \right]. \tag{12}$$

Note that in Eq. (12), $\mathbb{E}_{s_{t+1}} r(s_t, a_t, s_{t+1}) + \gamma V(s_{t+1}) \neq Q(s_t, a_t)$, because the transition $(s_t, a_t) \to s_{t+1}$ takes place in $\mathcal{P}'$ instead of $\mathcal{P}$. Now, we complete the proof of the relativity theory.

## B  PROOF OF THEOREM 2

Denote the dynamics-induced value gap as a function

$$\Delta^{\mathcal{P}', \mathcal{P}}(\pi) = J(\mathcal{P}', \pi) - J(\mathcal{P}, \pi)$$

$$= \mathbb{E}_{\tau \sim \mathcal{P}', \pi} \sum_{t=0}^{\infty} \gamma^t \left[ r(s_t, a_t, s_{t+1}) + \gamma V^{\mathcal{P}, \pi}(s_{t+1}) - Q^{\mathcal{P}, \pi}(s_t, a_t) \right].$$

We have

$$\Delta^{\mathcal{P}', \mathcal{P}}(\pi) - \Delta^{\mathcal{P}', \mathcal{P}}(\pi')$$

$$= \mathbb{E}_{\tau \sim \mathcal{P}', \pi} \sum_{t=0}^{\infty} \gamma^t \left[ r(s_t, a_t, s_{t+1}) + \gamma V^{\mathcal{P}, \pi}(s_{t+1}) - Q^{\mathcal{P}, \pi}(s_t, a_t) \right] -$$

$$\mathbb{E}_{\tau \sim \mathcal{P}', \pi'} \sum_{t=0}^{\infty} \gamma^t \left[ r(s_t, a_t, s_{t+1}) + \gamma V^{\mathcal{P}, \pi'}(s_{t+1}) - Q^{\mathcal{P}, \pi'}(s_t, a_t) \right].$$

Let

$$L_{\pi'}(\pi) = \sum_{t=0}^{\infty} \gamma^t \, \mathbb{E}_{s_0, a_0, \cdots, s_t \sim \mathcal{P}', \pi'} \sum_{a_t} \pi(a_t|s_t) \sum_{s_{t+1}} \mathcal{P}'(s_{t+1}|s_t, a_t)$$

$$\left[ r(s_t, a_t, s_{t+1}) + \gamma V^{\mathcal{P}, \pi'}(s_{t+1}) - Q^{\mathcal{P}, \pi'}(s_t, a_t) \right]$$

be the approximation of $\Delta^{\mathcal{P}',\mathcal{P}}(\pi)$ by sampling $(s_0, a_1, \cdots, s_t)$ and evaluating values using $\pi'$. Then, we have

$$\Delta^{\mathcal{P}',\mathcal{P}}(\pi) - L_{\pi'}(\pi)$$

$$= \mathbb{E}_{\tau \sim \mathcal{P}', \pi} \sum_{t=0}^{\infty} \gamma^t \left[ r(s_t, a_t, s_{t+1}) + \gamma V^{\mathcal{P},\pi}(s_{t+1}) - Q^{\mathcal{P},\pi}(s_t, a_t) \right] -$$

$$\mathbb{E}_{\tau \sim \mathcal{P}', \pi} \sum_{t=0}^{\infty} \gamma^t \left[ r(s_t, a_t, s_{t+1}) + \gamma V^{\mathcal{P},\pi'}(s_{t+1}) - Q^{\mathcal{P},\pi'}(s_t, a_t) \right] +$$

$$\mathbb{E}_{\tau \sim \mathcal{P}', \pi} \sum_{t=0}^{\infty} \gamma^t \left[ r(s_t, a_t, s_{t+1}) + \gamma V^{\mathcal{P},\pi'}(s_{t+1}) - Q^{\mathcal{P},\pi'}(s_t, a_t) \right] -$$

$$\sum_{t=0}^{\infty} \gamma^t \mathbb{E}_{s_0, a_0, \cdots, s_t \sim \mathcal{P}', \pi'} \sum_{a_t} \pi(a_t | s_t) \sum_{s_{t+1}} \mathcal{P}'(s_{t+1} | s_t, a_t)$$

$$\left[ r(s_t, a_t, s_{t+1}) + \gamma V^{\mathcal{P},\pi'}(s_{t+1}) - Q^{\mathcal{P},\pi'}(s_t, a_t) \right].$$

Let

$$D_1 = \mathbb{E}_{\tau \sim \mathcal{P}', \pi} \sum_{t=0}^{\infty} \gamma^t \left[ r(s_t, a_t, s_{t+1}) + \gamma V^{\mathcal{P},\pi}(s_{t+1}) - Q^{\mathcal{P},\pi}(s_t, a_t) \right] -$$

$$\mathbb{E}_{\tau \sim \mathcal{P}', \pi} \sum_{t=0}^{\infty} \gamma^t \left[ r(s_t, a_t, s_{t+1}) + \gamma V^{\mathcal{P},\pi'}(s_{t+1}) - Q^{\mathcal{P},\pi'}(s_t, a_t) \right],$$

and

$$D_2 = \mathbb{E}_{\tau \sim \mathcal{P}', \pi} \sum_{t=0}^{\infty} \gamma^t \left[ r(s_t, a_t, s_{t+1}) + \gamma V^{\mathcal{P},\pi'}(s_{t+1}) - Q^{\mathcal{P},\pi'}(s_t, a_t) \right] -$$

$$\sum_{t=0}^{\infty} \gamma^t \mathbb{E}_{s_0, a_0, \cdots, s_t \sim \mathcal{P}', \pi'} \sum_{a_t} \pi(a_t | s_t) \sum_{s_{t+1}} \mathcal{P}'(s_{t+1} | s_t, a_t)$$

$$\left[ r(s_t, a_t, s_{t+1}) + \gamma V^{\mathcal{P},\pi'}(s_{t+1}) - Q^{\mathcal{P},\pi'}(s_t, a_t) \right].$$

For $D_1$, we have

$$D_1 = \mathbb{E}_{\tau \sim \mathcal{P}', \pi} \sum_{t=0}^{\infty} \gamma^t \left[ \gamma \left( V^{\mathcal{P},\pi}(s_{t+1}) - V^{\mathcal{P},\pi'}(s_{t+1}) \right) - \right.$$

$$\left. \left( \mathbb{E}_{s_{t+1} \sim \mathcal{P}} \left[ r(s_t, a_t, s_{t+1}) + \gamma V^{\mathcal{P},\pi}(s_{t+1}) \right] - \mathbb{E}_{s_{t+1} \sim \mathcal{P}} \left[ r(s_t, a_t, s_{t+1}) + \gamma V^{\mathcal{P},\pi'}(s_{t+1}) \right] \right) \right]$$

$$= \sum_{t=0}^{\infty} \gamma^{t+1} \mathbb{E}_{s_0, \ldots, a_t \sim \mathcal{P}', \pi} \left[ \mathbb{E}_{s_{t+1} \sim \mathcal{P}'} \left( V^{\mathcal{P},\pi}(s_{t+1}) - V^{\mathcal{P},\pi'}(s_{t+1}) \right) - \mathbb{E}_{s_{t+1} \sim \mathcal{P}} \left( V^{\mathcal{P},\pi}(s_{t+1}) - V^{\mathcal{P},\pi'}(s_{t+1}) \right) \right]$$

$$= \sum_{t=0}^{\infty} \gamma^{t+1} \mathbb{E}_{s_0, \ldots, a_t \sim \mathcal{P}', \pi} \left[ \sum_{s_{t+1}} \left( \mathcal{P}'(s_{t+1} | s_t, a_t) - \mathcal{P}(s_{t+1} | s_t, a_t) \right) \left( V^{\mathcal{P},\pi}(s_{t+1}) - V^{\mathcal{P},\pi'}(s_{t+1}) \right) \right].$$

From the Lemma 1 in TRPO (Schulman et al., 2015), we have

$$|V^{\mathcal{P},\pi}(s_t) - V^{\mathcal{P},\pi'}(s_t)| = \mathbb{E}_{\tau \sim \mathcal{P}, \pi} \left[ \sum_{i=t}^{\infty} \gamma^{i-t} |A^{\pi'}(s_t, a_t)| \right] \leq \frac{\epsilon}{1 - \gamma}.$$

Therefore, we have

$$|D_1| \leq \sum_{t=0}^{\infty} \gamma^{t+1} \frac{2\delta_1 \epsilon}{1 - \gamma} = \frac{2\gamma \delta_1 \epsilon}{(1 - \gamma)^2}. \tag{13}$$

For $D_2$, we have

$$
\begin{aligned}
D_2 &= \sum_{t=0}^{\infty} \gamma^t \sum_{s_t} \left(p'_\pi(s_t) - p'_{\pi'}(s_t)\right) \sum_{a_t} \pi(a_t|s_t) \sum_{s_{t+1}} \\
&\quad \mathcal{P}'(s_{t+1}|s_t, a_t) \left[ r(s_t, a_t, s_{t+1}) + \gamma V^{\mathcal{P},\pi'}(s_{t+1}) - Q^{\mathcal{P},\pi'}(s_t, a_t) \right] \\
&= \sum_{t=0}^{\infty} \gamma^t \sum_{s_t} \left(p'_\pi(s_t) - p'_{\pi'}(s_t)\right) \sum_{a_t} \pi(a_t|s_t) \sum_{s_{t+1}} \\
&\quad \left(\mathcal{P}'(s_{t+1}|s_t, a_t) - \mathcal{P}(s_{t+1}|s_t, a_t)\right) \left( r(s_t, a_t, s_{t+1}) + \gamma V^{\mathcal{P},\pi'}(s_{t+1}) \right).
\end{aligned}
$$

Using the Lemma B.2 in MBPO, we have $D_{TV}(p'_\pi(s_t)||p'_{\pi'}(s_t)) \le t\delta_2$. Also, we can bound

$$
\left| r(s_t, a_t, s_{t+1}) + \gamma V^{\mathcal{P},\pi'}(s_{t+1}) \right| = \left| r(s_t, a_t, s_{t+1}) + \gamma \mathbb{E} \sum_{i=t+1}^{\infty} \gamma^{i-t-1} r(s_i, a_i, s_{i+1}) \right| \le \frac{r_{max}}{1-\gamma}.
$$

Finally, we have

$$
|D_2| \le \frac{4 r_{max} \delta_1 \delta_2}{1-\gamma} \sum_{t=0}^{\infty} t\gamma^t = \frac{4 r_{max} \delta_1 \delta_2 \gamma}{(1-\gamma)^3}. \tag{14}
$$

Combining Eqs. (13) and (14), we finally get the lower bound

$$
\Delta^{\mathcal{P}',\mathcal{P}}(\pi) \ge L_{\pi'}(\pi) - \frac{2\gamma \delta_1 \epsilon}{(1-\gamma)^2} - \frac{4 r_{max} \delta_1 \delta_2 \gamma}{(1-\gamma)^3}. \tag{15}
$$

Now, we complete the proof.

## C    PROOF OF PROPOSITION 1

Recall that $J(\mathcal{P}', \pi) = J(\mathcal{P}, \pi) + \Delta^{\mathcal{P}',\mathcal{P}}(\pi)$ and

$$
\begin{aligned}
L_{\pi'}(\pi) &= \sum_{t=0}^{\infty} \gamma^t \, \mathbb{E}_{s_0,a_0,\cdots,s_t \sim \mathcal{P}',\pi'} \sum_{a_t} \pi(a_t|s_t) \sum_{s_{t+1}} \mathcal{P}'(s_{t+1}|s_t, a_t) \\
&\quad \left[ r(s_t, a_t, s_{t+1}) + \gamma V^{\mathcal{P},\pi'}(s_{t+1}) - Q^{\mathcal{P},\pi'}(s_t, a_t) \right] \\
&= \sum_s d^{\mathcal{P}',\pi'}(s) \sum_a \pi(a|s) \sum_{s'} \mathcal{P}'(s'|s, a)[r(s, a, s') + \gamma V^{p,\pi'}(s') - Q^{p,\pi'}(s, a)] \\
&= \mathbb{E}_{s \sim d^{\mathcal{P}',\pi'}, a, s' \sim \mathcal{P}', \pi'} \frac{\pi(a|s)}{\pi'(a|s)} [r(s, a, s') + \gamma V^{\mathcal{P},\pi'}(s') - Q^{\mathcal{P},\pi'}(s, a)].
\end{aligned}
$$

Directly maximizing $L_{\pi'}(\pi)$ is insufficient, because once $\pi$ varies, $J(\mathcal{P}, \pi)$ varies (probably decreases) accordingly, which can not guarantee monotonic increasing in $J(\mathcal{P}', \pi)$. The case is that we have to maximize $J(\mathcal{P}, \pi) + \Delta^{\mathcal{P}',\mathcal{P}}(\pi)$. The policy improvement theorems in TRPO suggest

$$
J(\mathcal{P}, \pi) - J(\mathcal{P}, \pi') \ge \mathbb{E}_{\tau \sim \mathcal{P}, \pi'} \left[ \sum_{t=0}^{\infty} \gamma^t A^{\mathcal{P},\pi'}(s_t, a_t) \right] - \frac{4\epsilon \gamma \delta_2^2}{(1-\gamma)^2}.
$$

Therefore, we have the final bound

$$
\begin{aligned}
& J(\mathcal{P}', \pi) - J(\mathcal{P}, \pi') \\
&= J(\mathcal{P}', \pi) - J(\mathcal{P}, \pi) + J(\mathcal{P}, \pi) - J(\mathcal{P}, \pi') \\
&\geq L_{\pi'}(\pi) + \mathbb{E}_{\tau \sim \mathcal{P}, \pi'}\left[\sum_{t=0}^{\infty} \gamma^t A^{\mathcal{P},\pi'}(s_t, a_t)\right] - C \\
&= \mathbb{E}_{s \sim d^{\mathcal{P}',\pi'}, a, s' \sim \mathcal{P}', \pi'} \frac{\pi(a|s)}{\pi'(a|s)}[r(s,a,s') + \gamma V^{\mathcal{P},\pi'}(s') - Q^{\mathcal{P},\pi'}(s,a) + A^{\mathcal{P},\pi'}(s,a)] - C \\
&= \mathbb{E}_{s \sim d^{\mathcal{P}',\pi'}, a, s' \sim \mathcal{P}', \pi'} \frac{\pi(a|s)}{\pi'(a|s)}[r(s,a,s') + \gamma V^{p,\pi'}(s') - V^{p,\pi'}(s)] - C
\end{aligned}
\tag{16}
$$

where $C = \frac{2\gamma\delta_1\epsilon + 4\epsilon\gamma\delta_2^2}{(1-\gamma)^2} + \frac{4r_{max}\delta_1\delta_2\gamma}{(1-\gamma)^3}$. Now, we complete the proof.

## D  PROOF OF THEOREM 3

Continuing from Eq. (12), we have
$$J(\mathcal{P}', \pi) - J(\mathcal{P}, \pi)$$

$$
= \mathbb{E}_{\tau \sim \mathcal{P}'} \sum_{t=0}^{\infty} \gamma^t \left[r(s_t, a_t, s_{t+1}) + \gamma V^{\mathcal{P},\pi}(s_{t+1}) - Q^{\mathcal{P},\pi}(s_t, a_t)\right]
$$

$$
= \sum_{t=0}^{\infty} \gamma^t \mathbb{E}_{s_0, a_0, \cdots, a_t \sim \mathcal{P}', \pi} \left[\mathbb{E}_{s_{t+1} \sim \mathcal{P}'}\left(r(s_t, a_t, s_{t+1}) + \gamma V^{\mathcal{P},\pi}(s_{t+1})\right) - \mathbb{E}_{s_{t+1} \sim \mathcal{P}}\left(r(s_t, a_t, s_{t+1}) + \gamma V^{\mathcal{P},\pi}(s_{t+1})\right)\right]
$$

$$
= \sum_{t=0}^{\infty} \gamma^t \mathbb{E}_{s_0, a_0, \cdots, a_t \sim \mathcal{P}', \pi} \left[\sum_{s_{t+1}} \mathcal{P}'(s_{t+1}|s_t, a_t)\left(r(s_t, a_t, s_{t+1}) + \gamma V^{\mathcal{P},\pi}(s_{t+1})\right)\right.
$$

$$
\left. - \sum_{s_{t+1}} \mathcal{P}(s_{t+1}|s_t, a_t)\left(r(s_t, a_t, s_{t+1}) + \gamma V^{\mathcal{P},\pi}(s_{t+1})\right)\right].
$$

Now, considering a parameterized source dynamics $\mathcal{P}_\phi$, we have

$$
\begin{aligned}
\Delta^{\mathcal{P}',\mathcal{P}_\phi}(\pi) &= J(\mathcal{P}', \pi) - J(\mathcal{P}_\phi, \pi) \\
&= \sum_{t=0}^{\infty} \gamma^t \mathbb{E}_{s_0, a_0, \cdots, a_t \sim \mathcal{P}', \pi} \left[\sum_{s_{t+1}} \mathcal{P}'(s_{t+1}|s_t, a_t)\left(r(s_t, a_t, s_{t+1}) + \gamma V^{\mathcal{P}_\phi,\pi}(s_{t+1})\right)\right. \\
&\quad \left. - \sum_{s_{t+1}} \mathcal{P}_\phi(s_{t+1}|s_t, a_t)\left(r(s_t, a_t, s_{t+1}) + \gamma V^{\mathcal{P}_\phi,\pi}(s_{t+1})\right)\right].
\end{aligned}
$$

Let

$$
\begin{aligned}
L_{\phi'}(\phi) = & \\
\sum_{t=0}^{\infty} \gamma^t \mathbb{E}_{s_0, a_0, \cdots, a_t \sim \mathcal{P}', \pi} & \left[\sum_{s_{t+1}} \mathcal{P}'(s_{t+1}|s_t, a_t)\left(r(s_t, a_t, s_{t+1}) + \gamma V^{\mathcal{P}_{\phi'},\pi}(s_{t+1})\right)\right. \\
& \left. - \sum_{s_{t+1}} \mathcal{P}_\phi(s_{t+1}|s_t, a_t)\left(r(s_t, a_t, s_{t+1}) + \gamma V^{\mathcal{P}_{\phi'},\pi}(s_{t+1})\right)\right].
\end{aligned}
$$

Then,
$$\Delta^{\mathcal{P}',\mathcal{P}_\phi}(\pi) - L_{\phi'}(\phi) =$$

$$
\sum_{t=0}^{\infty} \gamma^{t+1} \mathbb{E}_{s_0, a_0, \cdots, a_t \sim \mathcal{P}', \pi} \left[\sum_{s_{t+1}} \left(\mathcal{P}'(s_{t+1}|s_t, a_t) - \mathcal{P}_\phi(s_{t+1}|s_t, a_t)\right)\left(V^{\mathcal{P}_\phi,\pi}(s_{t+1}) - V^{\mathcal{P}_{\phi'},\pi}(s_{t+1})\right)\right].
$$

For $V^{\mathcal{P}_\phi,\pi}(s_{t+1}) - V^{\mathcal{P}_{\phi'},\pi}(s_{t+1})$, we have

$$\left| V^{\mathcal{P}_\phi,\pi}(s_{t+1}) - V^{\mathcal{P}_{\phi'},\pi}(s_{t+1}) \right|$$

$$= \mathbb{E}_{\tau \sim \mathcal{P}_\phi,\pi} \sum_{i=t+1}^{\infty} \gamma^{i-t-1} \left| \sum_{s_{i+1}} \left( \mathcal{P}_\phi(s_{i+1}|s_i,a_i) - \mathcal{P}_{\phi'}(s_{i+1}|s_i,a_i) \right) \left( r(s_i,a_i,s_{i+1}) + \gamma V^{\mathcal{P}_{\phi'},\pi}(s_{i+1}) \right) \right|$$

$$\leq \frac{2\delta_1 r_{max}}{(1-\gamma)^2}.$$

Finally, we obtain the lower bound

$$|\Delta(\phi) - L_{\phi'}(\phi)| \leq \frac{4\delta_1^2 \gamma r_{max}}{(1-\gamma)^3}.$$

## E  THE RPO AND RTO ALGORITHMS

Due to space limitation, we are not able to put the RPO and RTO algorithms in the main text. These two algorithms can be tailored from the RPTO algorithm in Algorithm 1, and we put them here.

---

**Algorithm 2:** Relative Policy Optimization (RPO)

---

1. Give the source and target environments $\mathcal{E}^{source}$ and $\mathcal{E}^{target}$, and their dynamics $\mathcal{P}^{source}$ and $\mathcal{P}^{target}$; give a well-trained policy $\pi_{\theta_0}$ in $\mathcal{E}^{source}$;
2. Create two empty replay buffers $\mathcal{D}_{source}$ and $\mathcal{D}_{target}$;
3. Initialize $\theta = \theta_0$;
**while** *True* **do**

  4. Using $\pi_\theta$ to interact with $\mathcal{E}_\phi^{source}$ and push the generated trajectories into $\mathcal{D}_{source}$;
  5. Using $\pi_\theta$ to interact with $\mathcal{E}^{target}$ and push the generated trajectories into $\mathcal{D}_{target}$;
  6. Sample a mini-batch $\{(s,a,s')\}_{source} \sim \mathcal{D}_{source}$, and update $V^{\mathcal{P}_\phi^{source},\pi_\theta}$ by minimizing the TD-error;
  7. Sample a mini-batch $\{(s,a,s')\}_{target} \sim \mathcal{D}_{target}$, and apply the relative policy gradient in RPO to update $\pi_\theta$;
**end while**

---

---

**Algorithm 3:** Relative Transition Optimization (RTO)

---

1. Give the source and target environments $\mathcal{E}^{source}$ and $\mathcal{E}^{target}$, and their dynamics $\mathcal{P}_\phi^{source}$ and $\mathcal{P}^{target}$, where the source dynamics $\mathcal{P}_\phi^{source}$ is parameterized by $\phi$; give an arbitrary policy $\pi_\theta$;
2. Create two empty replay buffers $\mathcal{D}_{source}$ and $\mathcal{D}_{target}$;
3. Initialize $\phi = \phi_0$;
**while** *True* **do**

  4. Using $\pi_\theta$ to interact with $\mathcal{E}_\phi^{source}$ and push the generated trajectories into $\mathcal{D}_{source}$;
  5. Using $\pi_\theta$ to interact with $\mathcal{E}^{target}$ and push the generated trajectories into $\mathcal{D}_{target}$;
  6. Sample a mini-batch $\{(s,a,s')\}_{source} \sim \mathcal{D}_{source}$, and update $V^{\mathcal{P}_\phi^{source},\pi_\theta}$ by minimizing the TD-error;
  7. Sample a mini-batch $\{(s,a,s')\}_{target} \sim \mathcal{D}_{target}$, and update $\mathcal{P}_\phi^{source}$ according to RTO or SL;
**end while**

---

## F  OTHER EXPERIMENTAL DETAILS

For all the environments used in our experiments, the policy neural network and value neural network are of the same structure. The state is fed into three fully connected layers with ReLU activation function, and then the output embedding is fed into a policy head and a value head, respectively. That is, the policy network and value network share the bottom embeddings. For tasks with discrete action space, the policy head is a softmax layer to output multinomial distribution; for continuous control problems, the policy head is a diagonal Gaussian distribution that outputs a mean and a std.

The value head consists of two fully connected layers and finally outputs a scalar value. All the layers are of the same size of 64 in our experiments. For the physical dynamics model, each environment has its own physical systems coded in OpenAI gym. In CartPole-v0, only the pole length is treated as the trainable parameter, i.e., $\phi$ in CartPole-v0 only contains one free parameter. Similarly, in MountainCarContinuous-v0, only gravity is trainable; in Acrobot-v1, two physical factors link mass1 and link mass2 are trainable; in Pendulum-v0, the trainable parameter in $\phi$ is also the gravity.

For all algorithms, we choose their learning rate from [5e-4, 1e-4, 1e-5] with the best performance. For RPTO, its loss consists of the RPO loss and the RTO loss, and we simply fix both the loss coefficients as 1.0 in all experiments. For RPO, as mentioned in the main text, we clip the objective in Eq. (5) in a similar way as PPO did, and the clip range is set to 0.2, which is suggested by PPO.

For all the environments, we first pre-train a converged policy in the source environment using PPO-zero, and then use the trained parameters as warm start for all algorithms in the policy transfer experiments. However, for MountainCarContinuous-v0, PPO-zero fails to learn a good policy that can solve the source task. This phenomenon has also been observed in many previous approaches considering deep exploration, where MountainCarContinuous-v0 has been considered as a typical task for deep exploration. Nevertheless, it is not our focus in this paper, and as long as we can obtain a good policy in the source environment, this policy can be used as a warm start for the policy transfer stage, no matter what method is adopted to generate this policy. Therefore, to solve the source MountainCarContinuous-v0, we first handcraft a rule that can succeed in the source task, and then employ imitation learning to obtain a parameterized policy, which is further fine-tuned using PPO-zero in the source MountainCarContinuous-v0 environment. The final converged policy is used as the warm start at the policy transfer stage. For other environments, PPO-zero can provide a good policy in the source task.

