# OpenReview forum: "A General Theory of Relativity in Reinforcement Learning"
_ICLR.cc/2022/Conference — ICLR 2022 Submitted_

### Official Review · Reviewer_vBa4 · 2021-10-25

**Correctness:** 3
**Technical Novelty And Significance:** 1
**Empirical Novelty And Significance:** 1
**Recommendation:** 3
**Confidence:** 4

**Main Review:**

1. Equation (1),(4) is called the Performance Difference Lemma. Equation (3) is called the Simulation Lemma. See e.g. Lemma 1.16 and Lemma 2.2 of this textbook [https://rltheorybook.github.io/rltheorybook_AJKS.pdf]. These are standard tools for analyzing model-based RL, and you can also find them in almost every theoretical model-based RL paper, so I don't know what's surprising or new about Theorem 1...

2. Regarding the 1st remark below Theorem 1, the easiest way to estimate $J(P',\pi)-J(P,\pi)$ is to just run $\pi$ in both $P$ and $P'$... The variance of $J(P',\pi)$, thus the amount of data needed to estimate it accurately, is strictly smaller than the variance of the quantity inside the expectation on the RHS of equation (3).

3. The experimental results show only marginal improvement over PPO, especially since each iteration of RPO and RPTO collect twice the amount of data than PPO (one trajectory in $P$ and one trajectory $P'$).

**Summary Of The Paper:**

This paper proposes a model-based policy gradient algorithm that performs gradient updates on both the policy and a differentiable environment simmulator.

**Summary Of The Review:**

Overall, there is no theoretical justification of why the proposed method is better than vanilla TRPO/PPO, and the empirical advantage is also limited. The so-called \textit{theory of relativity} for RL is nothing but basic regret decompositions well-known in the RL community.

---

> ### Author Response · Authors · 2021-11-19
> **Response to Reviewer vBa4**
>
> **Question 1**: about the Performance Difference Lemma.
>
> **Answer**: Thanks for letting us know the existence of the Performance Difference Lemma. We were unaware of this and had biased assessment of our contribution. Please refer to the overall reply at the beginning.
>
> **Questions 2 and 3**: about the estimation of the dynamics-induced gap and the data usage in the experiments.
>
> **Answer**: Considering the two questions, we do not think you understand our main claims in this paper. As we have emphasized throughout the paper, we consider the problem of policy transfer in two MDPs, referred to as the source MDP and the target MDP. The data collection in the source MDP is assumed trivial and we can feel free to have infinite interactions with the source MDP efficiently, while we aim to reduce the interactions between the agent and the target environment. So, the easiest way of estimating the dynamics-induced gap by running $\pi$ in both $P$ and $P’$ is not considered immediately. Moreover, we are not focusing on directly estimating the dynamics-induced gap (as we have mentioned in the first paragraph in Section 4). Instead, we are seeking for algorithms that can incorporate data collection in both environments in one closed loop, and try to use the knowledge and data in the source task as much as possible while largely reducing the data collection in the target environment. Hence, in the experiments, we only compare the data collection in the target environment.

---

### Official Review · Reviewer_gbpy · 2021-11-01

**Correctness:** 2
**Technical Novelty And Significance:** 3
**Empirical Novelty And Significance:** Not applicable
**Recommendation:** 5
**Confidence:** 4

**Main Review:**

Strengths:
    1. The idea is novel and inspiring.
    2. The proposed algorithms are practical.
    3. The overall storyline is clear.

Weaknesses:
    1. Lot of confusions
        a) The main merit of RPO seems to be that RPO can collect infinite data from one MDP1 with which it only needs a few data from MDP2 to get a good policy in MDP2. However, it is not clear to me whether the merit was demonstrated from the experiments. How much data was actually used in each of the two MDPs? What is the meaning of "training step"? Do baseline algorithms use the same amount of data as RPO?
        b) I can not see how equation 7 is obtained from equation 6 in the deterministic case. Could you explain it?
        c) Could you explain the obtained bounds?
        d) The proposed algorithms (RPO and RTO) are derived from only parts of the bounds and ignore other parts of the bounds even though they are also relevant. And there is no explanation about why they can be ignored.
        e) pi_old and phi_old do not appear in the pseudo-code while they appear in the equations.
    2. Lots of experimental details are missing
        a) what is the optimizer?
        b) what is the initialization?
        c) what is PPO-zero?


**Summary Of The Paper:**

This paper proposed a way to decompose the difference between values of two policies in two MDPs respectively. Such a decomposition results in two parts, the first one is the difference between values of one policy under two MDPs; the second part is the difference between values of two policies under the same MDP. Using this decomposition, the paper then proposed three algorithms.

The first algorithm, called RPO, is used when there are two MDPs that the agent can interact with. The agent uses data from both two MDPs to get a good policy for one of the two MDPs. Such an algorithm is expected to be useful when gathering data from one MDP is costly while it from the other MDP is much cheaper.

The second algorithm is a model learning algorithm. In this setting, the agent only interacts with one MDP and learns a model to approximate the MDP. The RTO algorithm is different from the classic model learning algorithm (regression) in that the model is learned to achieve some consistency between the predicted values from the model and from the MDP.

The third algorithm combines the first two algorithms and is a full model-based algorithm. Specifically, now the agent only interacts with one MDP and learns a model of that MDP using RTO. Meanwhile, it also maintains and updates a policy that is expected to perform well in the MDP, using data from both the MDP and the model.



**Summary Of The Review:**

Overall I really like the main idea of this paper. But there are just so many things, including those that are key to their main results, that confuse me. I believe that this paper would become an important work to the field once it is clear.

---

> ### Author Response · Authors · 2021-11-19
> **Response to Reviewer gbpy**
>
> Thanks for the detailed comments.
>
> **Question**: “The main merit of RPO seems to be that RPO can collect infinite data from one MDP1 with which it only needs a few data from MDP2 to get a good policy in MDP2. However, it is not clear to me whether the merit was demonstrated from the experiments. How much data was actually used in each of the two MDPs? What is the meaning of ‘training step’?”
>
> **Answer**: In the experiments, in Fig. 1(a) and Fig. 2, the y-axis indicates the average return in the target environment (as depicted in these figures), and the ‘training step’ in the x-axis indicates the amount of training data from the target environment. Since we focus on the data usage in the target environment, we have omitted the statistics in the source environment. We will clarify this in future version.
>
> **Question**: “Do baseline algorithms use the same amount of data as RPO?”
>
> **Answer**: Yes, all the algorithms use the same amount of data in the target environment, and for these algorithms which need a warm start, they use the same initialization.
>
> **Question**: “I cannot see how equation 7 is obtained from equation 6 in the deterministic case. Could you explain it?”
>
> **Answer**: For deterministic case, we directly optimize the equation above Eq. (6), i.e.,
>
> $L_{\phi'}(\phi)=\sum_{t=0}^{\infty}\gamma^t\ E_{s_0,\cdots,a_t\sim P',\pi}\sum_{s_{t+1}}\left(P'(s_{t+1}|s_t,a_t)-P_{\phi}(s_{t+1}|s_t,a_t)\right)\left(r(s_t,a_t,s_{t+1})+\gamma V^{P_{\phi'},\pi}(s_{t+1})
> \right)$,
>
> which is Eq. (7) by substituting the next state $s_{t+1}$ with the deterministic dynamics model.
>
> **Question**: “Could you explain the obtained bounds?”
>
> **Answer**: The bounds in RPO and RTO are all derived for finding practical optimization objectives connecting the current parameters and the parameters since last update, because we require an iterative procedure to apply the gradients.
>
> **Question**: “The proposed algorithms (RPO and RTO) are derived from only parts of the bounds and ignore other parts of the bounds even though they are also relevant. And there is no explanation about why they can be ignored.”
>
> **Answer**: The bounds for RPO and RTO are independent except that both them are based on the main conclusion of the dynamics- and policy-induced gaps. The two algorithms can be applied independently. RPTO involves RPO and RTO in a closed-loop and the data usage of the two algorithms can be shared in RPTO.
>
> **Question**: “pi_old and phi_old do not appear in the pseudo-code while they appear in the equations.”
>
> **Answer**: pi_old and phi_old are parameters since last update, and the data are collected using the old parameters while we apply gradients for the current parameters.
>
> **Question**: “Lots of experimental details are missing a) what is the optimizer? b) what is the initialization? c) what is PPO-zero?”
>
> **Answer**: For all the experiments, we use the adam optimizer. For all the baselines, we initialize the policy using a pre-trained policy. Please refer to the second paragraph in the experimental section on page 8. PPO-zero indicates using PPO with random initialization.

---

### Official Review · Reviewer_G2Gj · 2021-11-07

**Correctness:** 3
**Technical Novelty And Significance:** 2
**Empirical Novelty And Significance:** 2
**Recommendation:** 3
**Confidence:** 2

**Main Review:**

Since my knowledge of empirical RL is very limited, below I will mostly comment from a theoretical perspective.

The first main theorem (Theorem 1) in this paper is already well-known and rather basic. It's a simple combination of performance difference lemma and model difference lemma.

The second main theorem (Theorem 2) provides the key theoretical justification for the two algorithms. Briefly speaking, it provides a low bound for the model-difference-induced gap that is relatively easier to optimize in practice. Its derivation is rather standard. And by only looking at the statement of this Theorem, this lower bound looks not very tight. The quantity on the LHS is at most $O(\delta_1)$ but the second term in the RHS is order $\Omega(\delta_1)$. Also, it's hard to tell whether the RHS could be negative in many cases. I am wondering in what cases this low bound would be a good approximation (either in the additive or multiplicative sense) to the quantity we want to maximize? Is there any theoretical justification?

In short, I am not well convinced by the theoretical justification in this paper.

Nevertheless, It's likely that it provides two really nice empirical algorithms that work pretty well in practice since I can see from the experimental section that the performance seems to be very nice. However, because I am not familiar with the standard benchmark tasks and algorithms in this field, I would like to hear more from other reviewers on this aspect.

**Summary Of The Paper:**

This paper studies how to fast adapt an old policy to a new but similar environment. It proposes two new algorithms RPO and RTO complemented with some theoretical justification and empirical performance demonstration.

**Summary Of The Review:**

See above.

---

> ### Author Response · Authors · 2021-11-19
> **Response to Reviewer G2Gj**
>
> Thanks for letting us know the existence of the main theory. In addition to the overall reply at the beginning, we answer other questions below.
>
> **Question**: “it's hard to tell whether the RHS could be negative in many cases” and theoretical justification of the tightness of the bound
>
> **Answer**: Thanks for the suggestion. We agree a theoretical justification of the tightness of the bound can provide better support for the proposed algorithms. We would like to elaborate on this in future version. The LHS of the bound in Theorem 2 is the dynamics-induced gap and itself is not necessarily positive, when the value in $P’$ is lower than that in $P$ given some $\pi$. Then, the RHS might be negative as well. This theorem tends to provide a lower bound that is possible to be optimized practically, similar to the approximation in TRPO.

---

### Official Review · Reviewer_U1Ad · 2021-11-08

**Correctness:** 3
**Technical Novelty And Significance:** 2
**Empirical Novelty And Significance:** 2
**Recommendation:** 3
**Confidence:** 4

**Main Review:**

This paper presents a grand-sounding “general theory of relativity in RL”, but both components of the theory are already known from prior work. The authors acknowledge that the explicit form of the policy-induced gap is given by the policy improvement theorem of Kakade and Langford, but they seem unaware that the explicit form of the dynamics-induced gap (Eq. 3) appeared in a 2018 paper [1] under the name “telescoping lemma” (Lemma 4.1). Considering that Eq. 3 is arguably the main novel (according to the authors) theoretical contribution of this submission, I think this is a serious issue, and the paper should not be accepted in its current form.

The algorithms are novel to my knowledge, however. I think a rewrite of the paper which just states Theorem 1 (acknowledging the prior work!) and then moves on to Theorems 2/3 and RPO/RTO/RPTO could still be a useful contribution. It would be stronger still if RPTO were shown to be effective for model-based RL, a setting which is frequently mentioned for motivation but not explored empirically.

The experimental results could be better. RPTO's performance looks good, but RPO fails to outperform PPO-warm on 2 out of 4 environments, and RTO’s performance is not even plotted.

Other comments on the writing:
* I feel pretty strongly that the phrase “general theory of relativity” should not be used because it already has a widely accepted usage in physics, leading to confusion from readers who go in expecting an interesting connection to or application of Einstein’s theory. A simple fix would be to replace all instances of “relativity” with “transfer”.
* “Instead, with much deeper investigation of the fundamental dynamics-induced value gap, we can finally get the explicit identity equation in Eq. (3), instead of a bound. As we will show later, Theorem 1 suggests two thoroughly new algorithms for policy transfer and transition update. Details of the superiority of Theorem 1 over MBPO can be found in the proofs in appendix.“ I understand that some degree of salesmanship is necessary in ML research these days, but I felt it should be toned down a bit in this passage.
* In Proposition 1, it is not correct to refer to $C$ as a constant, as it depends (via $\delta_2$) on the policy which you are trying to optimize.
* I do not understand why Eq. (6) follows from the previous equation. If you are using a non-trivial result from previous work, please cite it.

[1] Yuping Luo, Huazhe Xu, Yuanzhi Li, Yuandong Tian, Trevor Darrell, Tengyu Ma. Algorithmic Framework for Model-based Deep Reinforcement Learning with Theoretical Guarantees.

**Summary Of The Paper:**

The paper studies transfer in reinforcement learning (RL), beginning with a theorem that relates the performance of one policy under a particular dynamics to another policy under different dynamics. This is broken down into a “dynamics-induced gap” and a “policy-induced gap”, for which explicit expressions are given. Optimizing a bound on the policy-induced gap w.r.t. the policy leads to an algorithm they call Relative Policy Optimization (RPO), and similarly optimizing a bound on the dynamics-induced gap w.r.t. the dynamics leads to an algorithm they call Relative Transition Optimization (RTO). The two algorithms can be combined into a single algorithm, Relative Policy-Transition Optimization (RPTO), which optimizes both the policy and the dynamics. Experiments indicate that RPTO achieves better transfer performance, both in terms of sample efficiency and asymptotic performance, than RPO and PPO warm-started from an expert policy from the source task.

**Summary Of The Review:**

The submission fails to cite a previous paper which proved their main theoretical result, and the experiments leave room for improvement.

---

> ### Author Response · Authors · 2021-11-19
> **Response to Reviewer U1Ad**
>
> Thank you so much for letting us know the existence of the telescoping lemma. In addition to the overall reply above, we answer other questions below.
>
> **Question**: “I think a rewrite of the paper which just states Theorem 1 (acknowledging the prior work!) and then moves on to Theorems 2/3 and RPO/RTO/RPTO could still be a useful contribution.”
>
> **Answer**: Thanks for these positive comments on the proposed algorithms. We will carefully rewrite this manuscript and focus on our algorithms.
>
> **Question**: “The experimental results could be better. RPTO's performance looks good, but RPO fails to outperform PPO-warm on 2 out of 4 environments, and RTO’s performance is not even plotted.”
>
> **Answer**: From our motivation, we do not expect RPO can outperform PPO-warm, since it is only a component of RPTO. We plot RPO’s performance to show that RPO only works for the cases when the dynamics gap between the two MDPs are small. In the studied experiments, RTO always converges to the exact true dynamics, and we omitted their curves. We will add them in our revision.
>
> **Question**: about the phrase “general theory of relativity” and some sentences of salesmanship
>
> **Answer**: We put too much credit on the contribution of the dynamics-induced gap, which we thought is significant, and hence we used this term and some writing stuffs. We will rewrite this paper and these terms will not be used for sure.
>
> **Question**: “In Proposition 1, it is not correct to refer to C as a constant, as it depends (via $\delta_2$) on the policy which you are trying to optimize”
>
> **Answer**: This is true. We will correct it in the revision.
>
> **Question**: “I do not understand why Eq. (6) follows from the previous equation. If you are using a non-trivial result from previous work, please cite it.”
>
> **Answer**: We will supplement more details on this.

---

### Author Response · Authors · 2021-11-19
**Overall Reply**

**To all reviewers**,

We sincerely thank you for all your valuable comments, especially on letting us know the existence of the Telescoping Lemma (or Performance Difference Lemma). Indeed, we were unaware of this theoretical result until we read the review. Without of this knowledge, our assessment of the contribution of the proposed work is biased for sure. Based on this point, we completely accept the decisions from the reviews and appreciate the reviewers’ evaluations.

Despite this, we are still happy that we can investigate this dynamics-induced gap independently from fundamental derivations of the gap between the expected returns, which is somehow a different way comparing with the proof of the telescoping lemma which uses the telescoping sum. We are also happy to see that the reviewers acknowledge the novelty of the proposed algorithms.

Except the issue of the novelty of the main theory, for other questions and some misunderstanding of our work, we would like to provide explanations point-by-point below.

---

### Decision · Program_Chairs · 2022-01-20

**Decision:**

Reject

**Comment:**

The paper did not strike any reviewer as a critical addition to the literature, including various concerns regarding (1) the use of the general theory of relativity, (2) some components are well known in the past works.